



# The Earth Model Column Collaboratory (EMC²) v1.1: An Open-Source Ground-Based Lidar and Radar Instrument Simulator and Subcolumn Generator for Large-Scale Models

Israel Silber[1], Robert C. Jackson[2], Ann M. Fridlind[3], Andrew S. Ackerman[3], Scott Collis[2], Johannes Verlinde[1], and Jiachen Ding[4]

[1]Department of Meteorology and Atmospheric Science, Pennsylvania State University, University Park, PA, USA
[2]Envrionmental Sciences Divison, Argonne National Laboratory, Argonne, IL, USA
[3]NASA Goddard Institute for Space Studies, New York, NY, USA
[4]Department of Atmospheric Sciences, Texas A&M University, College Station, TX, USA

**Correspondence:** Israel Silber (ixs34@psu.edu)

**Abstract.**

Climate models are essential for our comprehensive understanding of Earth's atmosphere and can provide critical insights on future changes decades ahead. Because of these critical roles, today's climate models are continuously being developed and evaluated using constraining observations and measurements obtained by satellites, airborne, and ground-based instruments. Instrument simulators can provide a bridge between the measured or retrieved quantities and their sampling in models and field observations while considering instrument sensitivity limitations. Here we present the Earth Model Column Collaboratory (EMC²), an open-source ground-based lidar and radar instrument simulator and subcolumn generator, specifically designed for large-scale models, in particular climate models, but also applicable to high-resolution model output. EMC² provides a flexible framework enabling direct comparison of model output with ground-based observations, including generation of subcolumns that may statistically represent finer model spatial resolutions. In addition, EMC² emulates ground-based (and air- or space-borne) measurements while remaining faithful to large-scale models' physical assumptions implemented in their cloud or radiation schemes. The simulator uses either single particle or bulk particle size distribution lookup tables, depending on the selected scheme approach, to perform the forward calculations. To facilitate model evaluation, EMC² also includes three hydrometeor classification methods, namely, radar- and sounding-based cloud and precipitation detection and classification, lidar-based phase classification, and a Cloud Feedback Model Intercomparison Project Observational Simulator Package (COSP) lidar simulator emulator. The software is written in Python, is easy to use, and can be straightforwardly customized for different models, radars and lidars.

Following the description of the logic, functionality, features, and software structure of EMC², we present a case study of highly supercooled mixed-phase cloud based on measurements from the U.S. Department of Energy Atmospheric Radiation Measurement (ARM) West Antarctic Radiation Experiment (AWARE). We compare observations with the application of EMC² to outputs from four configurations of the NASA Goddard Institute for Space Studies (GISS) climate model (ModelE3) in single-column model (SCM) mode and from a large-eddy simulation (LES) model. We show that two of the four ModelE3





configurations can form and maintain highly supercooled precipitating cloud for several hours, consistent with observations and LES. While our focus is on one of these ModelE3 configurations, which performed slightly better in this case study, both of these configurations and the LES results post-processed with EMC$^2$ generally provide reasonable agreement with observed lidar and radar variables. As briefly demonstrated here, EMC$^2$ can provide a lightweight and flexible framework for comparing

the results of both large-scale and high-resolution models directly with observations, with relatively little overhead and multiple options for achieving consistency with model microphysical or radiation scheme physics.

## 1 Introduction

The representation of cloud processes in models is continuously advancing, conceptually, and in the level of details and complexity implemented in the micro- and macro-physical schemes (e.g., Lin et al., 2019; Cesana et al., 2019). These improvements

are reflected in the accuracy of the resulting model output (e.g., Klein et al., 2013; Lin et al., 2019; Myers et al., 2021; Wang et al., 2019), yet results still show large inter-model variability (e.g., Zelinka et al., 2020). This variability results from, among other sources, model weaknesses concerning atmospheric processes such as cloud geometrical and optical thicknesses (e.g., Cesana and Waliser, 2016; Klein et al., 2013), formation and transition of marine stratocumulus clouds (e.g., Rémillard and Tselioudis, 2015; Lin et al., 2014; Cesana et al., 2019), and the formation and maintenance of supercooled water (e.g., Cesana

et al., 2012; Tan and Storelvmo, 2016; Silber et al., 2019b).

Meaningful model evaluation benefits from a direct ("apples-to-apples") comparison with observations. For the evaluation of cloud representation, model output is often compared with active remote-sensing measurements from instrumentation such as lidars and radars, which provide information on the spatial structure of clouds and some direct indications about active microphysical processes. However, in these profiling cases, performing model evaluation is challenging because of observational

detectability constraints (e.g., signal extinction), and lack of retrievals of some microphysical quantities by these instruments, for example, hydrometeor number concentration or water content. In addition, spatial resolution differences between a model's output and an observing instrument's measurement resolution present an additional difficulty.

To bridge the gap between large-scale models such as weather or climate models and active remote-sensing observations, instrument simulators with different purposes have been developed to estimate observed parameters using model output. For

example, the Cloud-resolving model Radar SIMulator (CR-SIM; Oue et al., 2020) was developed to emulate zenith-pointing and scanning radar and lidar variables using high-resolution model output, with considerations of hydrometeor shape and the resulting scattering calculations (see also Mech et al., 2020). The Cloud Feedback Model Intercomparison Project Observational Simulator Package (COSP; Bodas-Salcedo et al., 2011; Swales et al., 2018), on the other hand, was developed to operate over large-scale model output targeting satellite data as observational constraints, although expansions for the emulation of

ground-based radars and lidars have been developed (e.g., Zhang et al., 2018; Kuma et al., 2020). Because of the demanding computation associated with the emulation of satellite measurements, COSP is typically implemented on-line into models' code to facilitate output.





To account for spatial resolution discrepancies, which are typically accentuated in the case of large-scale models due to their largely coarser resolution, some model evaluation studies and forward simulators emulate a higher spatial resolution by generating subcolumns (e.g., Bodas-Salcedo et al., 2008, 2011; Chepfer et al., 2008; Klein and Jakob, 1999; Lamer, 2019; Stephens et al., 2010; Swales et al., 2018; Webb et al., 2001). Statistics calculated using multiple generated subcolumns (faithful to the

processed model's physics) can be directly compared with the associated observations, thereby mitigating spatial resolution biases and errors.

Here we present the Earth Model Column Collaboratory (EMC$^2$), an open-source ground-based lidar and radar simulator and subcolumn generator, which is designed to operate over large-scale model output while being faithful to the physics implemented in models' microphysics or radiation schemes but can also be applied to high-resolution model output. The software

is written in Python, allowing quick installation and providing customizable operation (for scattering calculations, etc.). In section 2 we describe the subcolumn generator, the forward calculations, and classification routines currently implemented in EMC$^2$. Section 3 contains a brief description of the Python code and software. In section 4 we demonstrate its use by comparing observations of a case study of highly-supercooled drizzling cloud over West Antarctica (see Silber et al., 2019a) with application of EMC$^2$ to ouputs from a large-eddy simulation (LES) and the NASA Goddard Institute for Space Studies (GISS)

ModelE3 climate model (see Cesana et al., 2019) in single-column model (SCM) mode. Finally, section 5 provides a summary of the code features and case study demonstration.

## 2   EMC$^2$ Description

The following simulator description assumes large-scale model convective and/or stratiform cloud scheme outputs containing four hydrometeor classes: cloud water (cl), cloud ice (ci), rain (precipitating liquid; pl), and snow (precipitating ice; pi).

While these four hydrometeor classes are widely used in various microphysics schemes, we note that EMC$^2$ can be generally adapted to cases in which fewer or additional classes are used. We note that all the acronyms, abbreviations, and symbols used throughout this study are listed in Appendix A.

The subcolumn allocation and forward calculations in EMC$^2$ can be performed using three different approaches:

1. Radiation scheme approach, which largely treats hydrometeor fraction in a generalized manner and utilizes bulk (hy-

25       drometeor population) scattering lookup tables (LUTs) generated using specific size distribution assumptions.

2. Microphysics scheme approach, which includes full integration of single-particle scattering LUTs using hydrometeor particle size distributions prognosed by models with consideration of sub-grid hydrometeor class fraction variability assumptions.

3. Empirical non-bespoke approach, which implements empirical formulation from the literature in the forward calculations

30       and can be used with either of the hydrometeor fraction treatment methods (microphysics or radiation) in the subcolumn generator.





The bespoke radiation and microphysics approaches follow the assumptions and general logic implemented in large-scale model radiation and microphysics schemes, respectively. While our main focus throughout this manuscript will be on these two approaches in EMC², we will briefly present the empirical approach in sect. 2.4 and present some of its EMC²-processed output in sect. 4.

We note that the detailed description below of the radiation and microphysics approaches is congruent with the current implementation of these approaches in the GISS ModelE3 climate model. However, the core of these approaches is similar in other climate and Earth system models (ESMs), and EMC² can be easily adapted to fit specific variations in a model assumptions (see sect. 3). For example, the microphysics approach currently operates only on stratiform microphysics scheme output using a two-moment bulk scheme (Gettelman and Morrison, 2015, hereafter MG2) that has been implemented in climate

models such as the Community Earth System Model version 2 (CESM2) Community Atmosphere Model version 6 (CAM6) (Danabasoglu et al., 2020), the Energy Exascale Earth System Model (E3SM; Golaz et al., 2019), and ModelE3.

## 2.1    Allocation of Hydrometeors to Subcolumns

Prior to the radar and lidar forward calculations, EMC² generates subcolumns for each model output column. These subcolumns emulate a higher model spatial resolution, which partially reconciles the locality of ground-based measurements and

allows a more robust statistical model evaluation. Subcolumns are generated and populated with hydrometeors from the top-down with maximum random overlap between liquid and ice phases (henceforth, the overlap rule; see also Fan et al., 2011) using a similar approach to that described by Lamer (2019, ch. 6). EMC² translates hydrometeor fractions in the model grid to a binary set of hydrometeor-containing and hydrometeor-free subcolumn bins. That is, given a specified number of subcolumns ($N_s$; determined by the user), the total number of hydrometeor-filled subcolumn bins at model level $h$ and time step $t$ is equal to

the rounded value of $N_s \times f_{hyd}(h, t)$, where $f_{hyd}$ is the volume fraction of a hydrometeor class (e.g., $f_{cl}$, $f_{pi}$) or a generalized hydrometeor fraction ($f_{gen}$) used in the model radiation scheme, at the same model level and time step. Here and henceforth, we assume for simplicity an SCM output (no horizontal coordinate dimensions), even though EMC² can generally operate not only on SCM simulation output but also on global simulation output.

The following steps are applied in order to populate subcolumns (number $1, 2, ..., N_s$) with hydrometeors:

1. Convective cloud hydrometeors (cl and ci) are allocated to the first subcolumns (lowest index; if $f_{hyd}(h, t) > 0$), thereby generating cloud-containing subcolumns with maximum convective geometrical cloud depths.

2. Stratiform cloud hydrometeors are allocated to subcolumn bins unoccupied by convective cloud hydrometeors while noting that cl and ci are allocated simultaneously to consistently follow the overlap rule. At model level $h$, stratiform clouds are first randomly allocated to subcolumn bins with overlying stratiform clouds (at level $h+1$), followed, if

necessary, by random allocation to subcolumns with cloud-free bins directly above at level $h+1$. This order of processing where clouds are preferentially extended vertically conforms with assumptions that are often implemented in large-scale model radiative transfer calculations. In the case where overlying stratiform hydrometeors exist, the overlying layer





phase is not a factor of consideration, such that a subcolumn bin containing ci may be located right above a subcolumn bin containing cl or both cl and ci (mixed-phase), and vice versa.

3. Convective and stratiform precipitating hydrometeors (pl and pi) are allocated to subcolumns without convective-stratiform no-overlap restrictions, i.e., convective and stratiform precipitation may co-exist in a single subcolumn bin while complying with the overlap rule. Similar to the stratiform cloud allocation, precipitation is allocated with maximum random overlap. If some subcolumn bins are still to be populated with precipitating hydrometeors, these hydrometeors are randomly allocated to cloudy grid cells of the same type (convective or stratiform), followed by random allocation of hydrometeors to cloud-free subcolumn bins.

Once the subcolumns are populated with hydrometeors, per hydrometeor class except for stratiform cl in the case of the microphysics approach, hydrometeor mixing ratio is set in every hydrometeor-containing subcolumn bin by $q_{hyd}(s, h, t) = \bar{q}_{hyd}(h, t) / f_{hyd}(h, t)$, where $q_{hyd}$ and $\bar{q}_{hyd}$ designate the mixing ratio of a hydrometeor class (e.g., $q_{cl}$, $q_{ci}$) in subcolumn bin $s$ and a corresponding model grid cell mean, respectively. In the case of cl when using the microphysics approach, at every model level $h$ and time step $t$, $q_{cl}(s, h, t)$ is randomly set in cl-containing subcolumn bins such that it would comply with the sub-grid variability gamma distribution described by Morrison and Gettelman (2008, eq. 8) while adjusting the values in the last cl-containing subcolumn bin such that hydrometeor mass is conserved (as in the case of other hydrometeor classes), i.e.,

$$\frac{\sum_{i=1}^{N_s} q_{hyd}(i, h, t)}{N_s} = \frac{\bar{q}_{hyd}(h, t)}{f_{hyd}(h, t)}. \tag{1}$$

We note that sub-grid scale variability of cloud water in ModelE3 is tied to the sub-grid scale variability of moisture rather than set at a fixed value as in Morrison and Gettelman (2008).

In stratiform hydrometeor-containing subcolumn bins, hydrometeor number concentration is set for every hydrometeor class by $N_{hyd}(s, h, t) = \bar{N}_{hyd}(h, t) / f_{hyd}(h, t)$, where $N_{hyd}$ and $\bar{N}_{hyd}$ designate the number concentration of a hydrometeor class in subcolumn bin $s$ and a corresponding model grid cell, respectively. EMC$^2$ assumes that convective schemes do not diagnose $\bar{N}_{hyd}$, and hence, this information is currently not produced by the simulator.

## 2.2 Forward Calculation of Lidar Variables

### 2.2.1 Microphysics Approach

In the microphysics approach (currently applicable only to stratiform clouds), per hydrometeor diameter $D$, EMC$^2$ utilizes $q_{hyd}$ and $N_{hyd}$ to calculate the hydrometeor size distribution, $\phi_{hyd}(D, s, h, t)$, fully consistent with the MG2 scheme (see Morrison et al., 2009, eq. 1-3). Using LUTs containing full Mie calculations for spheres (following Bohren and Huffman, 1983, Appendix A) of single particle extinction and backscatter efficiencies at lidar operating wavelength $\lambda_l$ $\left(Q_{e_{hyd}}(D, \lambda_l)\right.$ and $Q_{bs_{hyd}}(D, \lambda_l)$, respectively), the lidar particulate extinction cross-section $\left(\alpha_{p_{hyd}}(s, h, t)\right)$ and backscatter cross-section



**Table 1.** Hydrometeor class parameter values implemented in EMC$^2$

| | cl | ci | pl | pi | Notes / references |
|---|---|---|---|---|---|
| Density [$kg/m^3$] | 1000 | 500 | 1000 | 250 | as in ModelE3 |
| Lidar ratio | 18 | 24 | 5.5 | 24 | Thorsen and Fu (2015, Table 3); Nott and Duck (2011) |
| Lidar linear depolarization ratio | 0.03 | 0.35 | 0.10 | 0.40 | Based on sect. 4's case; see Silber et al. (2019a) |
| $a$ ($b$) in terminal velocity power law | 3e-7 (2) | 700 (1) | 841.997 (0.8) | 11.72 (0.41) | $V = aD^b$; $a$ units m$^{1-b}$s$^{-1}$; cf. Morrison and Gettelman (2008, Table 2) |

$\left(\beta_{p_{hyd}}(s,h,t)\right)$ are calculated in every hydrometeor-bearing subcolumn bin by

$$\alpha_{p_{hyd}}(s,h,t) = \frac{\pi}{4}\int_{D_{min}}^{D_{max}}\phi_{hyd}(D,s,h,t)Q_{e_{hyd}}(D,\lambda_l)D^2dD \approx$$

$$\frac{\pi}{8}\sum_{i=1}^{N_D-1}\left(\phi_{hyd}(D_i,s,h,t)Q_{e_{hyd}}(D_i,\lambda_l)D_i^2 + \phi_{hyd}(D_{i+1},s,h,t)Q_{e_{hyd}}(D_{i+1},\lambda_l)D_{i+1}^2\right)\Delta D_{i,i+1} \tag{2a}$$

$$\beta_{p_{hyd}}(s,h,t) = \frac{\pi}{4}\int_{D_{min}}^{D_{max}}\phi_{hyd}(D,s,h,t)Q_{bs_{hyd}}(D,\lambda_l)D^2dD \approx$$

$$\frac{\pi}{8}\sum_{i=1}^{N_D-1}\left(\phi_{hyd}(D_i,s,h,t)Q_{bs_{hyd}}(D_i,\lambda_l)D_i^2 + \phi_{hyd}(D_{i+1},s,h,t)Q_{bs_{hyd}}(D_{i+1},\lambda_l)D_{i+1}^2\right)\Delta D_{i,i+1}, \tag{2b}$$

5    where we use the trapezoidal rule for discrete integration over a series of $D$ values, which can be unevenly spaced by $\Delta D_{i,i+1} = D_{i+1} - D_i$, while noting that $D_1 = D_{min}$ and $D_{N_D} = D_{max}$, where $N_D$ is the number of diameters for which $Q_{e_{hyd}}$, $Q_{bs_{hyd}}$, and $\phi_{hyd}$ are calculated. In the case of the Mie calculations currently available in EMC$^2$, $D_1 = 0.1\,\mu$m, $D_{N_D} = 1$ cm, and $\Delta D_{i,i+1}$ is constant and equals $0.1\,\mu$m. The complex refractive indices $\left(m_{hyd}(\lambda_l)\right)$ used for liquid hydrometeors in the Mie calculations can be taken from Segelstein (1981, Table 1) or Rowe et al. (2020, for a temperature of -10 °C). Refractive indices

10    for ice hydrometeors are taken from Warren and Brandt (2008). The Maxwell-Garnet equation (Bohren and Battan, 1980, eq. 1) for a mixture of ice and air is used to calculate the effective $m_{hyd}$ for ci and pi based on the ice densities implemented in EMC$^2$ for ModelE3 (Table 1) relative to bulk ice density of 917 kg/m$^3$.

The total $\alpha_p$ and $\beta_p$ $\left(\alpha_{p_{tot}}(s,h,t)\text{ and }\beta_{p_{tot}}(s,h,t)\text{, respectively}\right)$ are calculated as the sum of each of these variables for cl, ci, pl, and pi. The lidar linear depolarization ratio (LDR) is estimated by weighting fixed LDR values (per hydrometeor class;

15    see Table 1) with the relative contribution of $\beta_{p_{hyd}}$ to $\beta_{p_{tot}}$.

The cumulative optical thickness (from the surface upward) at the base of a given subcolumn bin, $\tau_{hyd}$, is calculated by

$$\tau_{hyd}(s,h,t) = \sum_{i=2}^{h}\alpha_{p_{hyd}}(s,i-1,t)\Delta z(i-1,t), \tag{3}$$



where $\Delta z(h,t)$ denotes the geometrical thickness of model level $h$ at time step $t$ and $\tau_{hyd}(s,h=1,t)=0$. The total integrated optical thickness, $\tau_{tot}(s,h,t)$, being the sum of $\tau_{hyd}$ for cl, ci, pl, and pi, is used to estimate the level of full lidar signal attenuation (received signal not detectable by the simulated instrument), the value of which can be used to constrain comparisons between model output forward calculations and observations. Lidar signal extinction at visible wavelengths typically occurs

at an optical thickness of 3-5 (e.g., Sokolowsky et al., 2020, fig. 4), and hence, EMC[2] assumes by default that the lidar signal is extinct at a level where $\tau_{tot}=4$. We note that EMC[2] allows calculating $\tau_{hyd}$ from the top-down (i.e., at the top of a given model layer), thereby enabling simulation of airborne and spaceborne lidar measurements.

In some cases, the observational dataset only consists of measurements made by elastic lidars that do not allow direct retrieval of $\alpha_p$ and $\beta_p$. In these cases, observations can be compared with a diagnostic attenuated $\beta_{p_{tot}}$, $\beta_{p_{tot,att}}$, which is often referred

to as the normalized relative backscatter (NRB; Campbell et al., 2002) after several lidar signal corrections are applied. $\beta_{p_{tot,att}}$ is calculated here by

$$\beta_{p_{tot,att}}(s,h,t) = [\beta_p(s,h,t) + \beta_m(s,h,t)]T_m^2(s,h,t)\exp^{-2\eta\tau_{tot}(s,h,t)} \tag{4}$$

where $\beta_m$ and $T_m^2$ are the molecular backscatter cross-section and the two-way transmittance calculated following Penndorf (1957), and $\eta$ is the multiple scattering coefficient, the value of which is assumed to be equal to 1 by default, but can be

manually set to other fixed values based on the physical assumptions made or certain empirical results (e.g., Winker, 2003).

### 2.2.2  Radiation Approach

With the radiation approach, applicable to both stratiform and convective cloud scheme output, forward calculations rely on bulk scattering LUTs. Therefore, this approach is more than two orders of magnitude faster than the microphysics approach, thereby rendering EMC[2] more suitable for the analysis of large model output datasets. Using this approach, a geometric cross-

sectional area for each hydrometeor-bearing subcolumn bin is first calculated assuming geometric scatterers:

$$A_{hyd}(s,h,t) = \frac{3}{4}\frac{q_{hyd}(s,h,t)\rho_a(h,t)}{\rho_b r_{e_{hyd}}(h,t)}, \tag{5}$$

where $\rho_a$ is the density of air, $\rho_b$ is the bulk density of the hydrometeor class phase (1000 and 917 $kg/m^3$ is the case of liquid and solid water, respectively), and $r_{e_{hyd}}$ is the effective radius of a hydrometeor class in the model grid cell. The $\alpha_{p_{hyd}}$ and $\beta_{p_{hyd}}$ are then calculated by

$$\alpha_{p_{hyd}}(s,h,t) = Q_{e,vol_{hyd}}\left(r^*_{e_{hyd}}(h,t),\lambda_l\right)A_{hyd}(s,h,t) \tag{6a}$$

$$\beta_{p_{hyd}}(s,h,t) = Q_{bs,vol_{hyd}}\left(r^*_{e_{hyd}}(h,t),\lambda_l\right)A_{hyd}(s,h,t), \tag{6b}$$

where $Q_{e,vol_{hyd}}(\lambda_l)$ and $Q_{bs,vol_{hyd}}(\lambda_l)$ represent in this case bulk efficiencies per unit volume taken from LUTs, in which they are provided as function of $r_{e_{hyd}}$. In Eq. 6a and 6b, however, $Q_{e,vol_{hyd}}$ and $Q_{bs,vol_{hyd}}$ are functions of the adjusted effective





radius, $r^*_{e_{hyd}}$, which equals $r_{e_{hyd}}$ in all hydrometeor classes and cloud types except for stratiform cloud ice and snow. In these two cases, $r^*_{e_{hyd}} = r_{e_{hyd}} \Phi^*_{hyd}$ and $\Phi^*_{hyd} = \Phi_{hyd} \frac{\rho_{hyd}}{\rho_b} + (1 - \Phi_{hyd})(\frac{\rho_{hyd}}{\rho_b})^{\frac{1}{3}}$, where $\rho_{hyd}$ is the hydrometeor class density (see Table 1), and $\Phi_{hyd}$ is a constant fluffiness factor. $\Phi_{hyd}$ is used such that a value of 0 represents an equivalent mass bulk sphere as in Gettelman and Morrison (2015) while a value of 1 represents a fluffy sphere with an equivalent maximum dimension. In the case of ModelE3, for example, $\Phi_{hyd}$ is set by default to an intermediate value of 0.5, and generally serves as one of many tuning parameters

The default $Q_{e,vol_{hyd}}$ and $Q_{bs,vol_{hyd}}$ in eq. 6a and 6b (respectively) implemented in EMC$^2$ were calculated using single particle full Mie calculations in the case of liquid hydrometeors, and single particle scattering LUTs for a severely roughened 8-column ice aggregate (Yang et al., 2013) in the case of solid hydrometeors. These ice aggregate scattering calculations have been shown by Holz et al. (2016) to enable a closure between infrared Moderate-Resolution Imaging Spectroradiometer (MODIS; Platnick et al., 2003) and visible Cloud-Aerosol Lidar with Orthogonal Polarization (CALIOP; Winker et al., 2009) satellite ice optical thickness retrievals, and were included in the MODIS collection 6 (C6) cloud product (Platnick et al., 2017). In order to calculate the $Q_{e,vol_{hyd}}$ and $Q_{bs,vol_{hyd}}$ LUTs, we assumed the same gamma distribution parameters as those implemented in the C6 dataset (see Hansen, 1971, eq. 1), consistent with the bulk LUTs utilized by ModelE3's radiation scheme.

Following the calculations of $\alpha_{p_{hyd}}$ and $\beta_{p_{hyd}}$, the total variables $\alpha_{p_{tot}}$ and $\beta_{p_{tot}}$ as well as $\tau_{hyd}$, $\tau_{tot}$, and $\beta_{p_{tot,att}}$ are calculated as in the microphysics approach (sect. 2.2.1).

## 2.3 Forward Calculation of Radar Variables

### 2.3.1 Microphysics Approach

When the microphysics approach is selected in EMC$^2$, the first three radar moments are calculated for each hydrometeor class in every hydrometeor-bearing subcolumn bin; that is, the equivalent reflectivity factor ($Z_{e_{hyd}}$), the mean Doppler velocity ($V_{D_{hyd}}$) and the Doppler spectral width ($\sigma_{D_{hyd}}$) as well as total radar moment variables $\left(Z_{e_{tot}}, V_{D_{tot}}, \text{ and } \sigma_{D_{tot}}\right)$. Full Mie calculation LUTs for the emulated radars (Table 2) are first used to calculate $\beta_{p_{hyd}}$ at the radar operating wavelength $\lambda_r$ following eq. 2b. The $m_{hyd}(\lambda_r)$ used for liquid in the Mie calculations can be taken from Segelstein (1981, Table 1) or Turner et al. (2016, using a temperature of -10 °C), while $m_{hyd}(\lambda_r)$ values for ice are taken from Mätzler (2006, ch. 5.3; using temperature of -10 °C). $Z_{e_{hyd}}$ in every hydrometeor-containing subcolumn bin is then calculated (in linear units) using (Doviak and Zrnić, 1993, eq. 4.33)

$$Z_{e_{hyd}}(s,h,t) = \frac{\beta_{p_{hyd}}(s,h,t)\lambda_r^4}{\pi^5|K_w|^2},\tag{7}$$

where $|K_w|^2$ is the dielectric factor for water used in the raw radar observational processing (see Table 2). Using the resultant $Z_{e_{hyd}}$, $V_{D_{hyd}}$ is then calculated by implementing the hydrometeor class terminal velocities parametrization used in the MG2 scheme (cf. Morrison and Gettelman, 2008, Table 2). In the calculation of $V_{D_{hyd}}$ we neglect the model grid cell vertical wind,





**Table 2.** Radar instruments and some of their characteristics currently implemented in EMC$^2$. The ARM SGP, ENA, NSA, AWR, and MOS site abbreviations denote the Southern Great Plains, Eastern North Atlantic, North Slope of Alaska, the AWARE campaign (at McMurdo Station, Antarctica), and the Multidisciplinary Drifting Observatory for the Study of Arctic Climate (MOSAiC) Expedition, respectively. The calculations of the minimum detectable signal ($Z_{e_{min}}$) for KAZR are based on the analysis in Silber et al. (2018a). All $Z_{e_{min}}$ values correspond to 2 $s$ integration time except for the AWR and MOS WACR, the values of which correspond to 0.2 $s$ integration time.

| Parameter and ARM site | | XSACR | KAZR | WACR | Sources / references |
|---|---|---|---|---|---|
| Frequency [GHz] | | 9.71 | 34.86 | 95.04 | Widener and Johnson (2006); Widener et al. (2012a, b) |
| Index of refraction for water ($|K_w|^2$) | | 0.93 | 0.88 | 0.84 | Widener and Johnson (2006); Widener et al. (2012a, b) |
| $Z_{e_{min}}$ at 1 km [dBZ] | SGP | N/A | -51.5 | -46.0 | 8 year KAZR dataset analysis; Widener and Mead (2004) |
| | ENA | N/A | -56.5 | N/A | Analysis of 3.5 year KAZR2 dataset analysis |
| | NSA | N/A | -48.5 | N/A | 7.5 year KAZR dataset analysis |
| | AWR | -30.0 | -45.5 | -40.0 | Falconi et al. (2018); 1 year KAZR dataset analysis; Burns et al. (2016) |
| | MOS | -30.0 | -41.6 | -40.0 | Falconi et al. (2018); 1 year KAZR dataset analysis; Burns et al. (2016) |

$w$, which predominantly has little impact on the $V_{D_{hyd}}$ value. Finally, $\sigma_{D_{hyd}}$ is calculated, while we note that beamwidth and turbulent broadening (e.g., Chen et al., 2018) are omitted from this calculation, but will be added in future work.

To allow a valid comparison between the forward calculations and observations, signal attenuation is considered in the calculation of the attenuated $Z_{e_{tot}}$, $Z_{e_{tot,att}}$:

$$Z_{e_{tot,att}}(s,h,t) = 10log_{10}\big(Z_{e_{tot}}(s,h,t)\big) - 2\big(Y_{hyd_{tot}}(s,h,t) + Y_{gas}(s,h,t)\big). \tag{8}$$

where $Y_{gas}$ and $Y_{hyd_{tot}}$ are the one-way integrated attenuation at the base of the subcolumn bin (in dB) due to atmospheric gases ($O_2$ and $H_2O$; see Ulaby et al., 1981, sect. 5.3-5.5) and all hydrometeors, respectively. $Y_{hyd_{tot}}$ is calculated using

$$Y_{hyd_{tot}}(s,h,t) = 10log_{10}(e)\sum_{i=2}^{h}\alpha_{p_{tot}}(s,i-1,t)\Delta z(i-1,t), \tag{9}$$

where $\alpha_{p_{tot}}$ is determined by summing $\alpha_{p_{hyd}}$ based on eq. 2a calculated at $\lambda_r$ over all hydrometeor classes while setting $Y_{hyd_{tot}}(s,h=1,t) = 0$.

The vertical profile of the minimum detectable equivalent reflectivity factor, $Z_{e_{min}}$, is calculated by

$$Z_{e_{min}}(h,t) = Z_{e_{min}}(z=1000\,m) + 20log_{10}(z(h,t)/1000), \tag{10}$$

where $z(h,t)$ is the height at the base of model level $h$ (in meters) at time step $t$ and $Z_{e_{min}}(z=1000\,m)$ is the minimum detectable signal at 1 km (using the highest sensitivity mode; Table 2). When compared with observations, subcolumn bins where $Z_{e_{tot,att}}(s,h,t) < Z_{e_{min}}(h,t)$ can be treated as returned signal below the radar noise floor, and hence, are effectively considered hydrometeor-free.





### 2.3.2 Radiation Approach

When the radiation approach is selected in EMC$^2$, forward radar calculations using bulk LUTs are limited to the zeroth radar moment ($Z_e$) due to a set of limitations:

1. Large-scale model radiation schemes are not informed with hydrometeor fall velocities. Moreover, fall velocity parametrizations in microphysics schemes do not necessarily fully overlap with the hydrometeor size and shape assumptions implemented in radiation schemes.

2. Noting that EMC$^2$ operates off-line, hydrometeor class fall velocities are typically reported in model outputs as weighted means. Because not all cloud schemes enable back-tracing of hydrometeor class fall velocities using analytical expressions and weighted output fields (e.g., the convective cloud scheme in ModelE3; see Elsaesser et al., 2017), hydrometeor class fall velocities per subcolumn bin cannot be straightforwardly reproduced.

3. The total radar moments include combinations of the different hydrometeor class mixing ratios, and hence, cannot be determined using a single set of bulk LUTs per hydrometeor class.

Thus, EMC$^2$ calculates only the $Z_{e_{hyd}}$, $Z_{e_{tot}}$, and $Z_{e_{tot,att}}$. Eqs. 6b and 7 are used to calculate $Z_{e_{hyd}}$ with bulk scattering LUTs for $\lambda_r$ based on full single particle Mie scattering calculations in the case of liquid hydrometeors and single particle scattering calculations for the Yang et al. (2013) 8-column aggregate at a temperature of 270 K (see Ding et al., 2017) in the case of solid hydrometeors. Similar to the implementation of the radiation approach in the forward lidar calculations (sect. 2.2.2), $r_{e_{hyd}}$ considers the fluffiness factor in the case of solid (ice) hydrometeors. Finally, $Z_{e_{tot}}$, and $Z_{e_{tot,att}}$ are calculated similar to the microphysics approach using eq. 8 and 9 with $\alpha_{p_{hyd}}$ (and $\alpha_{p_{tot}}$) calculated using eq. 6a.

## 2.4 Empirical Forward Calculation Approach

Similar to the radiation approach, the empirical approach can be applied to both convective and stratiform cloud scheme outputs, and is limited to the zeroth moment in the case of forward calculations of radar variables. With this approach, empirical relationships between water content and instrument variables are used, similar to the forward simulator described by Lamer et al. (2018, Table 2). The lidar $\alpha_{p_{hyd}}$ is calculated in hydrometeor-bearing subcolumn bins using the formulation in Heymsfield et al. (2014, eq. 9 and 9d) and Hu et al. (2007, eq. 3) for ice and liquid hydrometeors, respectively, both of which were derived based on lidar measurements at $\lambda_l = 532\ nm$. The lidar $\beta_{p_{hyd}}$ is then calculated using the retrieved $\alpha_{p_{hyd}}$ while assuming a constant lidar ratio (the extinction to backscatter ratio) per hydrometeor class (see Table 1). We note that lidar operating wavelengths are not considered in these forward calculations due to the scarcity and the limited ability to validate these empirical parametrizations for different regions and cloud regimes.

In the forward radar calculations, $Z_{e_{hyd}}$ is estimated for the cloud water and rain hydrometeor classes using Fox and Illingworth (1997, eq. 11) and Hagen and Yuter (2003, sect. 4), respectively. Here we implicitly assume that the reflectivity factor ($Z$) is equivalent to $Z_e$, and hence, the radar wavelength has no weight over the calculation. $Z_{e_{hyd}}$ for cloud ice and snow is





estimated with the parametrization in Hogan et al. (2006, Table 2) using different parameters for the Ka and W radar bands. For the calculation of $Z_{e_{tot,att}}$, $Y_{hyd_{tot}}$ at the model level base is estimated (in dB) following Doviak and Zrnić (1993, eq. 3.17c):

$$Y_{hyd_{tot}}(s,h,t) \approx 10log_{10}(e)Im(-K_m)\sum_{i=1}^{h}\frac{6\pi\text{LWC}(s,i-1,t)}{\rho_w\lambda_r}\Delta z(i-1,t),\tag{11}$$

where $\rho_w = 1000\,kg/m^3$ is the water density, $K_m = (m_{hyd}^2-1)/(m_{hyd}^2+2)$, $LWC$ designates the sum of the cloud water and

precipitating liquid (rain) water content (in $g/m^3$), and $Y_{hyd_{tot}}(s,h=1,t) = 0$. Signal attenuation due to ice hydrometeors is neglected here due to its typical significantly weaker influence relative to liquid hydrometeors, as well as the large uncertainties associated with the irregular shape of ice particles (see Doviak and Zrnić, 1993, ch. 3.3).

## 2.5 Hydrometeor Classifications

Once the total lidar and/or radar variables are calculated, EMC$^2$ can be used to classify the subcolumn simulator output.

Currently, EMC$^2$ incorporates three hydrometeor classification methods: the radar-sounding cloud and precipitation detection and classification, the modified fixed lidar variable threshold phase classification, and the COSP lidar simulator emulator (henceforth referred to as the COSP emulator).

The radar-sounding cloud and precipitation detection method (Silber et al., 2021a; see also Vassel et al., 2019) emulates the combined use of relative humidity with respect to water (RH) sounding measurements for the detection of liquid-bearing

clouds and radar echoes (received signal above the radar noise floor) for the detection of precipitating hydrometeors. In the case of large-scale model output, RH below 100% in a model grid cell does not necessarily indicate a lack of cloud water, because of implemented assumptions concerning the sub-grid distribution of cloud water content (e.g., Smith, 1990). Therefore, the approach most consistent with the observational method is to simply use the cloud water-bearing subcolumn bins (see sect. 2.1) to classify the subcolumn bin as "cloud". "Precipitation" are those subcolumns bins in which $Z_{e_{tot,att}} \geq Z_{e_{min}}$. Subcolumn

bins that can be classified as both "cloud" and "precipitation" are set as "mixed". We note that at temperatures below 0 °C, the "mixed" classification type becomes more likely to represent a mixed-phase cloud with decreasing temperatures, but in general, may represent bins containing only liquid hydrometeors.

The modified fixed lidar variable threshold phase classification method is similar to previous studies that incorporated fixed LDR and $\beta_{p_{tot}}$ thresholds to classify hydrometeor-bearing air volumes to "liquid" and "ice-only" using lidar measurements

(e.g., Shupe, 2007; Thorsen and Fu, 2015). By default, however, EMC$^2$ includes two additional "undefined" classes that cover intermediate regions in the LDR-$\beta_{p_{tot}}$, such that a subcolumn bin classified as "undefined1" has a higher probability that it includes some amount of liquid water while "undefined2" is more likely to only contain ice hydrometers (see sect. 4.3 for discussion and illustration of the default thresholds). The notion behind the addition of these two "undefined" classes is the fixed-threshold method limitations that could originate in:

1. Drizzle- or rain-bearing air-volumes, which may produce moderate $\beta_{p_{tot}}$ and LDR on the order of 0.1, especially when ice hydrometeors are present in the same air volumes (e.g., Derr et al., 1976; Sassen, 2003; Silber et al., 2019a).





2. Cases in which an entire relatively tenuous liquid-bearing cloud layer or other cases with liquid-bearing air-volumes just above cloud base occur concurrently with ice precipitation sufficiently concentrated and intense to generate a combination of high $\beta_{p_{tot}}$ and LDR that can reach values of 0.10-0.20 (e.g., Derr et al., 1976; Silber et al., 2018b, fig. S1). This influence of ice hydrometeors also applies to rain-bearing air volumes.

3. Horizontally-oriented ice hydrometeors that may produce low (moderate) LDR ($\beta_{p_{tot}}$) via specular reflection even in cases when the lidar is titled up to several degrees off-zenith (e.g., Noel et al., 2002; Silber et al., 2018b, Appendix A). Note that this limitation only applies to observations, since specular reflection and ice particle canting angles are not represented in large-scale models.

Adaptive fixed thresholds that vary with site, instrument, and period of study (e.g., Silber et al., 2018a; Zaremba et al., 2020)
or lidar ratio constraints (e.g., Thorsen and Fu, 2015) can compensate for some of these limitations. However, these approaches cannot be objectively translated to the model output domain to enable a direct comparison between the observations and the simulator output. Therefore, the modified fixed threshold routine, which largely agrees with existing measurements yet acknowledges both model and observational uncertainties may allow better comparisons to be made.

The emulator of the COSP lidar simulator follows the same equations and logic of the on-line lidar simulator (Cesana and
Chepfer, 2013) implemented in numerous climate models. Note that, unlike the on-line COSP lidar simulator, this emulator operates using the model vertical levels and does not interpolate the model output onto an evenly-spaced vertical grid. As with all other EMC[2] forward calculations and classification routines, this emulator can operate using a top-down viewing approach thus providing a bridge between the COSP simulator and EMC[2].

Finally, EMC[2] also includes internal functions to calculate mass or frequency phase ratios using each of these hydrometeor
classification methods, providing metrics to compare model output with observations or with outputs from other models.

## 3   Software Description

EMC[2] depicts a workflow for comparing forward calculated radar or lidar variables generated from large-scale model output with radar or lidar measurements. Fig. 1 shows a flowchart example of this workflow for using EMC[2] to compare ModelE3's output with high spectral resolution lidar (HSRL) measurements. The workflow starts with the `Model` class incorporated
within the `emc2.core` module. The `Model` class contains model output field namelists and default hydrometeor parameters (Table 1). Using Python's class inheritance, EMC[2] allows the creation of a custom class specifying a given model's namelists and parameters (a `ModelE3` class in this example), which ensures that the model output can be standardized and used by the other modules in EMC[2]. Once loaded through the `Model` class internal methods, model output data are stored within the `Model` object using the `xarray` dataset module (Hoyer and Hamman, 2017).

The `Model` object is then input to the subcolumn generator (sect. 2.1). The results of the subcolumn generator are stored in the `xarray` dataset contained within the `Model` object. Here, we introduce the `Instrument` class. Similar to the `Model`



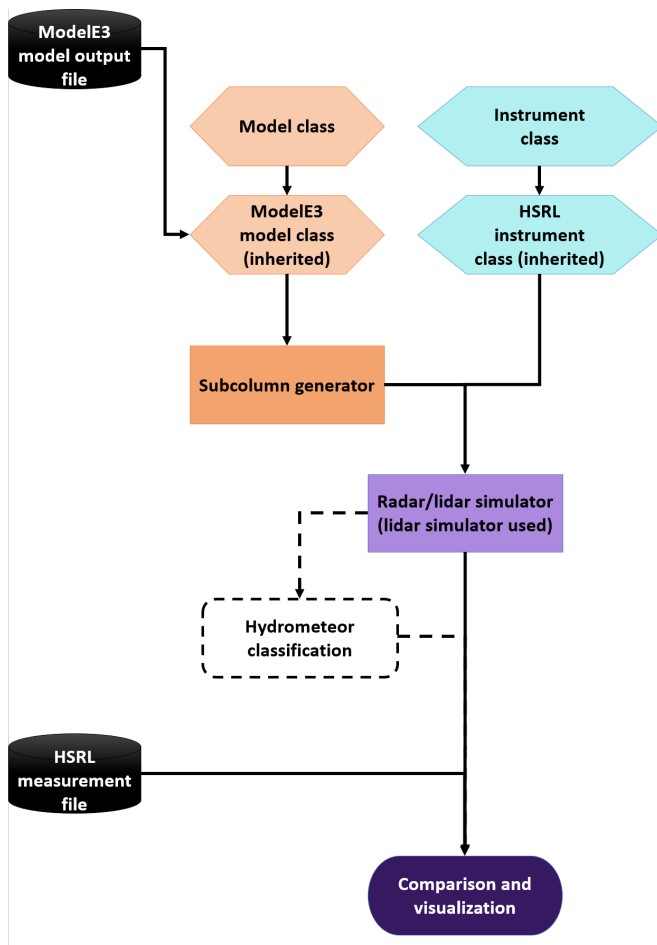

**Figure 1.** Standard workflow of EMC$^2$ utilization for direct comparison between observations and model output. The flowchart exemplifies the use of EMC$^2$ to compare the NASA GISS ModelE3 climate model output with corresponding high spectral resolution lidar (HSRL) measurements.

class, the `Instrument` class contains relevant information about the instrument being simulated (some of which is listed in Table 2) as well as the single-particle and bulk scattering calculation LUTs. Currently, zenith-pointing instrument properties and scattering calculation LUTs are available for various lidars and radars operated by the Department of Energy (DOE) Atmospheric Radiation Measurement (ARM) climate research facility. That is, the elastic micropulse lidar (MPL) and HSRL,

5 both operating at a wavelength of 532 nm (e.g., Flynn et al., 2007; Eloranta, 2005), the 910 nm CL31 ceilometer (Morris, 2016), the 1064 nm HSRL elastic channel (see Razenkov and Eloranta, 2018), the elastic channel of ARM's Raman lidar operating at 355 nm (e.g., Newsom, 2009), the X-band scanning ARM cloud radar (XSACR; Widener et al., 2012b), the Ka-band ARM zenith radar (KAZR; Widener et al., 2012a), and the W-band ARM cloud radar (WACR and M-WACR; Widener and Johnson, 2006). Similar to the `Model` class, the `Instrument` class allows EMC$^2$ to be tailored to other radars and lidars deployed at





different sites, which does not confine the analysis of measurements and model output to specific ARM instruments or sites, as long as the required parameters and suitable scattering LUTs are provided. Thus, the various variables mentioned in the previous section (e.g., $\Phi_{hyd}$, $\rho_{hyd}$, LDR) as well as the LUTs, can be easily specified and set to match configurations and assumptions implemented in different large-scale models, as well as complex scattering models more commonly implemented
in cloud-resolving and LES models.

Following the subcolumn generator process, the `Instrument` and `Model` objects are then input to the lidar (sect. 2.2) or radar (sect. 2.3) simulators (lidar simulator in the case of fig. 1). The forward calculation results are stored in the same `xarray` dataset in the `Model` object. Simulated hydrometeor classification (sect. 2.5) can be performed following the completion of the forward calculations and stored in the `xarray` dataset. For comparison and visualization of these results, EMC[2] uses the
Atmospheric Community Toolkit (ACT; Theisen et al., 2020). Thus, a `SubcolumnDisplay` object, inherited from ACT's `Display` object contains the necessary methods for quick visualization of the simulated instrument variables. In addition, the `SubcolumnDisplay` object also contains several internal methods for generating curtain and profile plots of observational and simulated data stored in the `Instrument` or `Model` objects, allowing masking of simulated signals below instrument detectability, for example. The figures generated in the next section (sect. 4) show examples of EMC[2]'s visualization capabili-
ties. Finally, since the data are in the `xarray` dataset format, EMC[2] also contains all of `xarray`'s analysis and visualization capabilities for these simulated datasets.

EMC[2] incorporates a suite of unit tests for each function using the `pytest` testing tool (https://pytext.readthedocs.io/) to inspect the integrity and functionality of the code. These unit tests are combined with continuous code integration using TravisCI integration service (https://travis-ci.com/), which runs the unit tests every time a developer submits a pull request on
GitHub. If the unit test passes with the developer's changes to the code, then the changes are approved to be a part of EMC[2]. Documentation is also automatically generated from the metadata strings in each subroutine to ensure that each part of the code is well documented.

## 4    Case Study Example: Highly Supercooled Antarctic Cloud

To demonstrate the application of EMC[2] and its output using the different forward calculation approaches, here we describe
and analyze a Lagrangian LES case study (Silber et al., 2019a) adjusted for running and testing the ModelE3 climate model (as well as other climate models) in SCM mode.

### 4.1    Case Description

As described by Silber et al. (2019a), the stratiform cloud event that we compare with model simulations was observed over
McMurdo Station, Antarctica, as part of the ARM West Antarctic Radiation Experiment (AWARE; Lubin et al., 2020), on August 16, 2016. During the event, cyclone-driven wind confluence with southwesterly katabatic flow resulted in relatively warm and moist marine air convergence along the Ross Ice Shelf coast, part of which was advected towards the McMurdo

Station measurement site (fig. 2a). This air convergence induced a widespread (>1000 km) cloud field evident by Clouds and the Earth's Radiant Energy System (CERES) measurements indicating an extensive region with enhanced top of atmosphere (TOA) upwelling longwave radiation (dashed green shape in fig. 2b; note that a surface-based temperature inversion is common during the austral winter resulting in smaller TOA radiation fluxes).

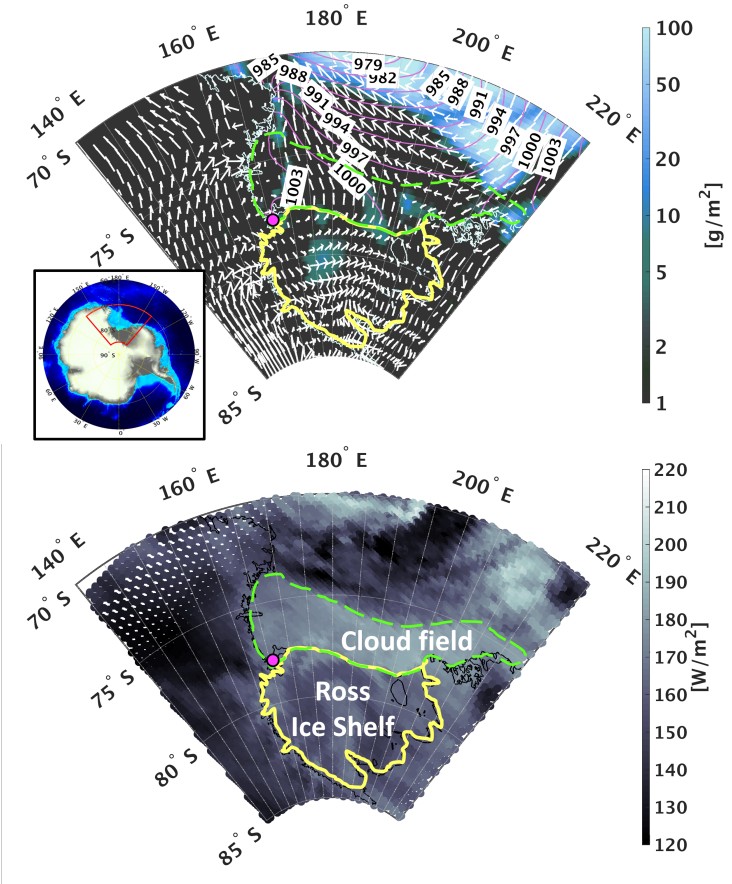

**Figure 2.** (Top) Sea level pressure (contours; land areas are masked), total condensate water path (logarithmic color-scale), and surface winds (quivers) resolved by ERA5 (Hersbach et al., 2020) for August 16, 2016, at 10:00-11:00 UTC. The magenta-filled marker denotes McMurdo Station. The dashed green shape highlights a widespread cloud field along the Ross Ice Shelf (yellow shape) coast associated with the ground-based observations. The inset panel shows a topographic map of Antarctica (the red box highlights the region depicted in the main panel). (Bottom) Top of atmosphere upwelling longwave radiation measured by CERES Aqua on August 16, 2016, at 10:44-10:50 UTC. The 1 arc-minute topographic data were developed by the National Geophysical Data Center (NGDC Amante, 2009) and are freely available at https://www.ngdc.noaa.gov/mgg/global/global.html. Reproduced from Silber et al. (2019a).





Over McMurdo Station, a decoupled persistent mixed-phase cloud with temperatures as low as -29 °C was observed for ~39 hours. The observed cloud was nearly continuously precipitating ice particles and was also drizzling for more than 7 hours, concluded from a comprehensive analysis of sounding, HSRL, and KAZR measurements (see Silber et al., 2019a).

## 4.2 ModelE3 SCM Configuration

Based on Lagrangian simulations constrained by the remote-sensing observations, Silber et al. (2019a) postulated that the activated ice nucleating particle (INP) and cloud condensation nuclei (CCN) concentrations during the event were on the order of 0.2 $L^{-1}$ and 20 $cm^{-3}$, respectively, to enable drizzle to be produced and precipitate along with ice precipitation below the highly supercooled cloud base. Their simulations were 9 h in duration, starting on August 16, 2016, at 01:00 UTC and ending at 10:00 UTC. By imposing large-scale vertical wind extracted from back-trajectory calculations, they emulated the

transport of the cloud layer, initially forming in a stable layer, towards McMurdo Station. The end of the simulation at 10:00 UTC designated the time at which the cloud field reached the fixed observational site at McMurdo, and hence, statistics of that hour of observations (10:00-11:00 UTC) were compared with the model output.

Here, we slightly adjusted the case study initialization files to enable running this case in a climate model's SCM mode, while using the same single-hour of radar and lidar observations and the Distributed Hydrodynamic Aerosol and Radiative Modeling

Application (DHARMA) model (Stevens et al., 2002) baseline LES output (see Silber et al., 2019a) as benchmarks. Namely, we simplified the profiles of vertical motion (as in Silber et al., 2020) and then converted the height coordinates of the initial sounding (see Silber et al., 2019a) and vertical wind time series to pressure coordinates (these converted sounding and forcing files are available at http://dx.doi.org/10.17632/gz4gdn3jvz.1). The utilization of these files to run the highly supercooled cloud case study enables testing of an SCM, and hence, the cloud schemes implemented in a climate model.

The LES is initialized here with activated INP concentration of 0.1 $L^{-1}$ and with both cloud ice and snow hydrometeor classes (whereas Silber et al. (2019a) used only a single ice class), and the simplified vertical motion profiles are then imposed. These few LES adjustments make the model configuration more consistent with climate model microphysics and SCM initialization while effectively resulting in the same hydrometeor content and cloud evolution as the baseline simulation presented in Silber et al. (2019a) (not shown). Thus, the SCM is run equivalently to the LES; that is, using the same initial sounding

and forcing files, setting the coordinates to -77.85 °S, 166.72 °E, and prescribing a monomodal log-normal aerosol particle concentration of 20 $cm^{-3}$ with a mean radius of 0.1 $\mu m$, geometrical standard deviation of 2, and a hygroscopicity parameter of 0.4. We note that activated INP concentrations are not prescribed in the SCM simulations because (a) ModelE3's final configurations are defined by specific values of certain model tuning parameters (among others) associated with INP parameterization, and (b) diagnostically prescribing the activated INP concentration is not faithful to the temperature-dependent

approach implemented in the model, and hence, would not necessarily be informative of true climate model weaknesses.





### 4.3 Comparison Between Observations and ModelE3 SCM Using EMC$^2$

The following figures show the EMC$^2$ forward calculation results using the DHARMA LES simulation and ModelE3's configuration Tun3, one of four configurations of ModelE3 derived in part via a machine learning parameter tuning approach that will be described in a manuscript in preparation, to be included in the Coupled Model Intercomparison Project Phase 6 (CMIP6).

The SCM using this configuration was able to generate a cloud-top inversion and turbulent layer via cloud-top radiative cooling, and produced the best agreement with the observations and the LES relative to the other three configurations (see Appendix B). Whereas here we examine application of EMC$^2$ to a single ModelE3 configuration in SCM mode in a case where we can also compare with LES, we note that EMC$^2$ is designed to enable detailed evaluation of atmospheric thermodynamic profile and cloud properties extracted from global simulations of ModelE3 configurations and other climate models against long-term

datasets at fixed sites in future dedicated work.

The left panels in Fig. 3 show the mixing ratios of the four hydrometeor classes evolving through the simulated SCM case study. These depicted mixing ratios are the output of the SCM's stratiform cloud scheme, but because this simulation does not generate any convective hydrometeors (as expected), they also represent the total mixing ratios. Cloud water mass (top panel) dominates over the other hydrometeor classes in hydrometeor-bearing model grid cells through much of the simulation, even

at lower levels in which the cloud water fraction is rather low (fig. 3, right panels). Rain (effectively drizzle) is produced by the model as well but has a relatively smaller mass compared with that of snow generated. This reduced amount of rainwater in the SCM simulation relative to rainwater dominance in the simulations of Silber et al. (2019a) is largely the result of the different autoconversion parameterization schemes implemented in ModelE3 (Seifert and Beheng, 2001) relative to that implemented by default in the DHARMA LES (Khairoutdinov and Kogan, 2000), which produces significantly smaller rain mass mixing

ratios in this case (not shown), contrary to some previous studies (e.g., Heiblum et al., 2016; Xiao et al., 2021). Understanding the source of this differing autoconversion parameterization behavior requires a dedicated study that is a beyond the scope of this manuscript.

Figs. 4 and 5 depict the HSRL and KAZR variables observed during a single hour over McMurdo Station and simulated

with EMC$^2$ using the DHARMA LES three-dimensional output at the end of that simulation (without using the subcolumn generator), and by applying each of the three approaches on ModelE3 configuration Tun3 SCM output for 05:00 UTC. Because our goal in this section is to demonstrate that EMC$^2$ can reasonably match cloud observations given comparable input, we present the EMC$^2$-processed SCM output 4 hours into the simulation when cloud top heights are similar to observed instead of the end of the simulation (the SCM develops the supercooled cloud faster than the baseline LES; see Appendix B). When

evaluating the processed model output against the observations, we essentially exchange temporal resolution with spatial resolution (three-dimensional model domain in the case of the LES) or an emulated spatial resolution (in the case of the SCM). We set the number of SCM subcolumns ($N_s$) to 100, which effectively represents hydrometeor class fractions up to the second decimal point, and enables drawing more robust statistics by emulating sub-grid variability of all hydrometeor classes combined.



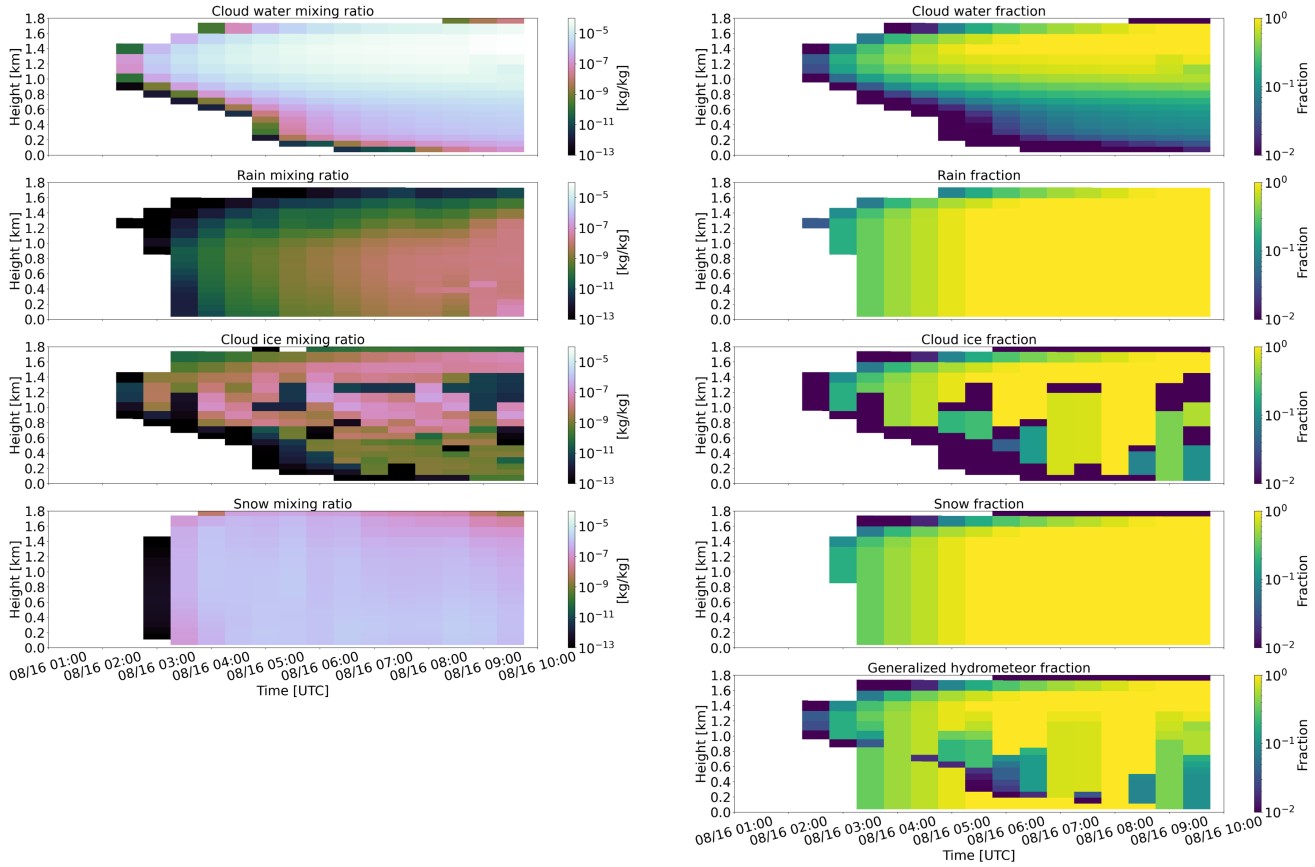

**Figure 3.** ModelE3 configuration Tun3 single column model (SCM) output for the Antarctic cloud case study (August 16, 2016) showing mixing ratio (left) and hydrometeor fraction (right) time-height curtain plots of (from top to bottom) cloud water, rain, cloud ice, and snow. The generalized hydrometeor fraction used in the radiation approach (see sec. 2) is depicted in the bottom right panel.

The processed LES output exhibits generally good agreement with the observations, evident by the comparable lidar and radar variable values and their horizontal variability (Figs. 4 and 5), the vertical cloud structure and boundaries, as well as the

5   full lidar signal attenuation near cloud top (Fig. 4). A multi-layer cloud structure developed by the LES is suggested by the intermittent breaks in the large $\beta_{p_{tot}}$ and $\alpha_{p_{tot}}$ values. This multi-layer structure is also indicated by the lidar observations (Fig. 4), and was comprehensively discussed by Silber et al. (2019a).

Using the microphysics approach, the SCM sub-grid variability is more pronounced relative to the radiation approach owing to the occurrence of cloud water combined with the implementation of its sub-grid variability as defined in the MG2 micro-

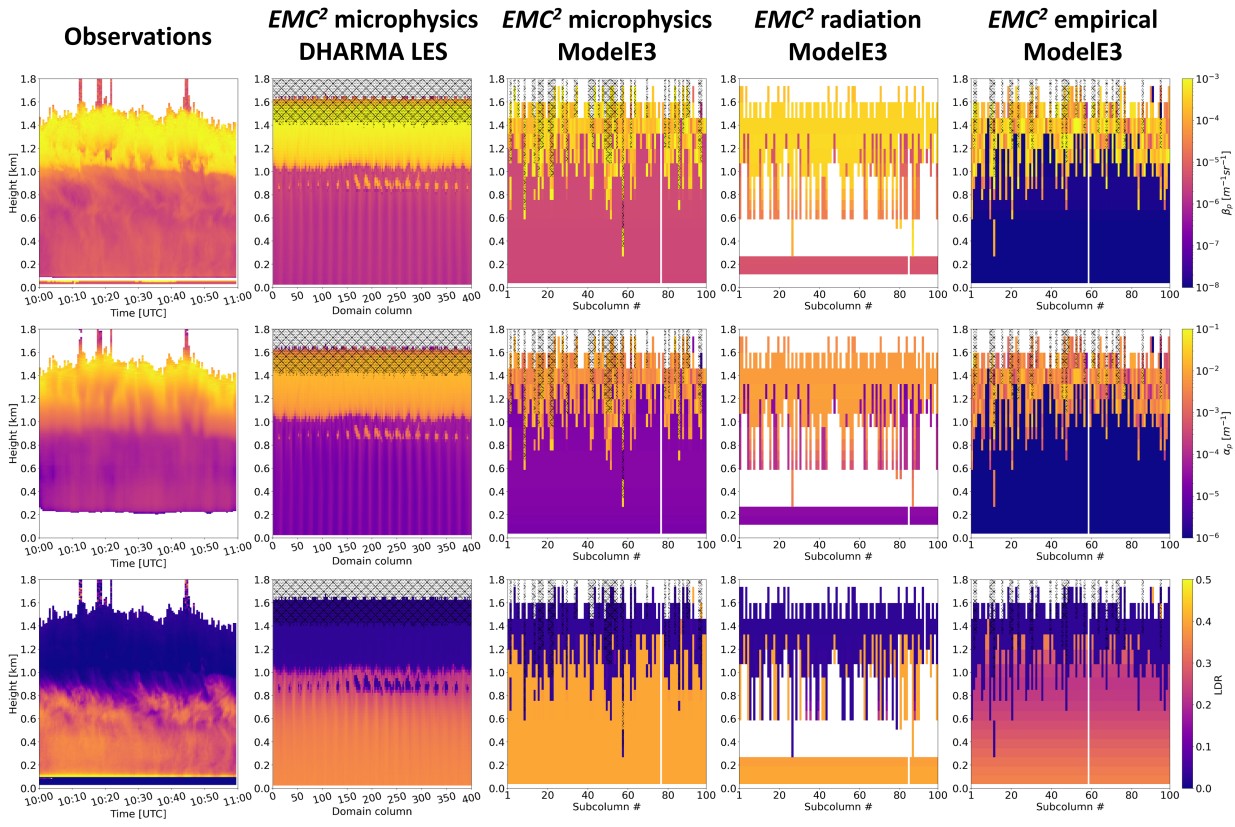

**Figure 4.** Observed and simulated HSRL particulate backscatter cross-section ($\beta_{p_{tot}}$; top row), particulate extinction cross-section ($\alpha_{p_{tot}}$; middle row), and linear depolarization ratio (LDR; bottom row) using the DHARMA LES output and ModelE3 configuration Tun3 SCM output. The columns show (from left to right) the observations from McMurdo Station between 10:00-11:00 UTC, DHARMA LES three-dimensional output processed using EMC$^2$ microphysics (without the subcolumn generator), ModelE3 EMC$^2$ output using the microphysics approach, EMC$^2$ output using the radiation approach, and EMC$^2$ output using the empirical approach with the microphysics cloud fractions for the subcolumn generator (see sect. 2.1). The DHARMA LES output corresponds to 10:00 UTC (arrival of cloud field at McMurdo Station; see sect. 4.2), while ModelE3's EMC$^2$ panels depict the SCM output for 05:00 UTC (see text) processed with EMC$^2$ using 100 subcolumns. A full lidar signal attenuation mask of $\tau_{tot} > 4$ ($\tau_{tot}$ is the total accumulated optical thickness) is applied to the plotted simulated data (hatched areas).

physics scheme. Evaluation of heights with full cloud cover, indicated by the large $\beta_{p_{tot}}$ and $\alpha_{p_{tot}}$ values (see also fig. 3, right), shows that the SCM is in reasonable agreement with the observations there. ModelE3's macrophysics scheme leads to notably ragged liquid cloud base heights that are evident in the SCM fields compared to the more uniform liquid cloud base in both observations and LES, as discussed further below. Relatively enhanced $\alpha_{p_{tot}}$ values are produced in some SCM subcolumns,



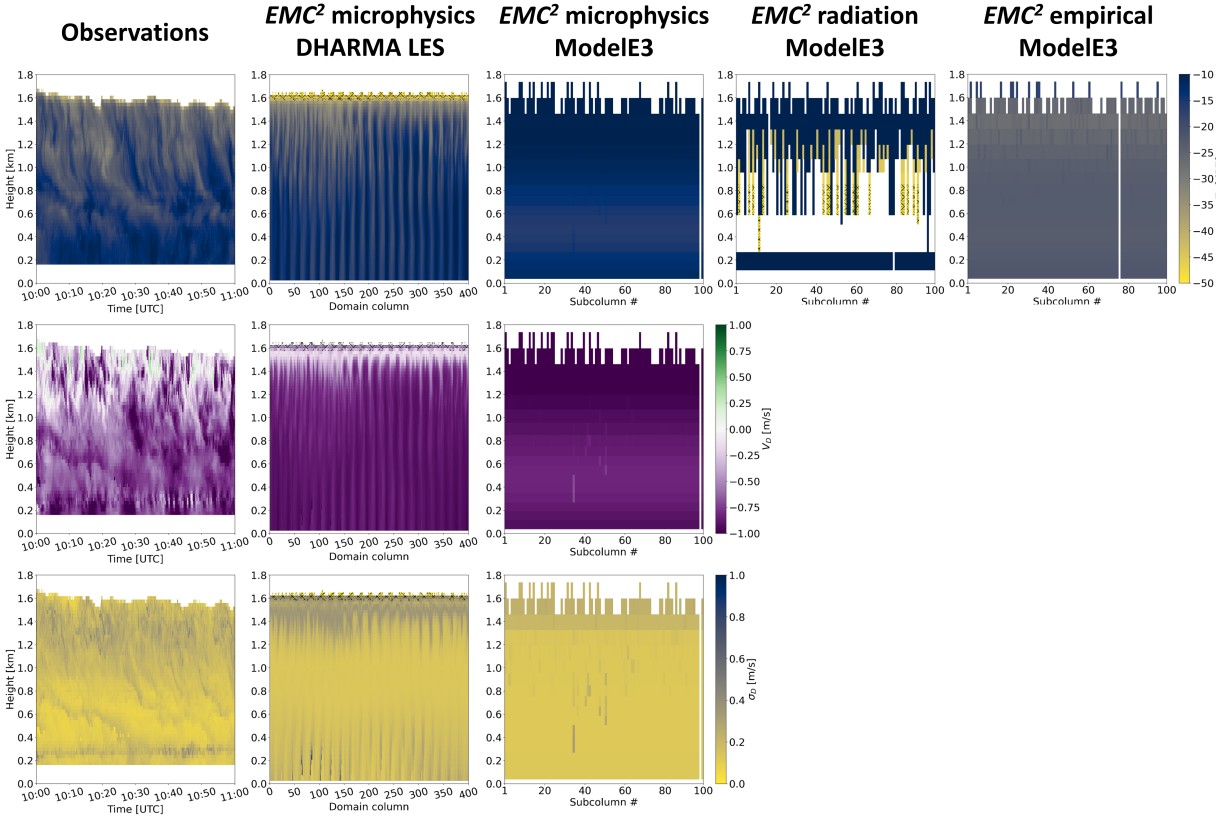

**Figure 5.** As in fig. 4, but with the KAZR attenuated equivalent reflectivity factor ($Z_{e_{tot,att}}$; top row), mean Doppler velocity ($V_{D_{tot}}$; middle row), and spectrum width ($\sigma_{D_{tot}}$; bottom row). Note that only $Z_{e_{tot,att}}$ is calculated in EMC²'s radiation and empirical approaches (see sects. 2.3.2 and 2.4). A radar signal-to-noise ratio (SNR) mask of $Z_{e_{min}} > Z_{e_{tot}}$ ($Z_{e_{min}}$ is the minimum detectable signal on each model level) is applied to the plotted data (hatched areas).

which leads to full lidar signal attenuation, as also indicated by the HSRL measurements (fig. 4). Full lidar signal attenuation does not occur in this case using the radiation approach because of the uniform distribution of cloud water between subcolumns commensurate with $\tau_{tot}$ values just below the full attenuation threshold of 4 (not shown). The subcolumns with full lidar signal attenuation in the microphysics approach call for the use of radar measurements for cloud-top detection (e.g., fig. 5 discussed below).

Except for the overestimated LDR values below the supercooled cloud layer relative to the observations, the values of the other two lidar variables and the general scenario structure appear to agree with the observations and LES in both the microphysics and radiation approaches, which exhibit an encouraging consistency with each other (fig. 4). The comparison of





the subcolumn-averaged lidar-variable profiles illustrated in fig. 6 allows a more quantitative comparison. This figure better indicates that the DHARMA LES $\alpha_{p_{tot}}$ within the main liquid-bearing cloud layer is within the range of the observations and their uncertainty, consistent with the conclusions of Silber et al. (2019a) that activated CCN number concentrations were low during this highly supercooled drizzle event.

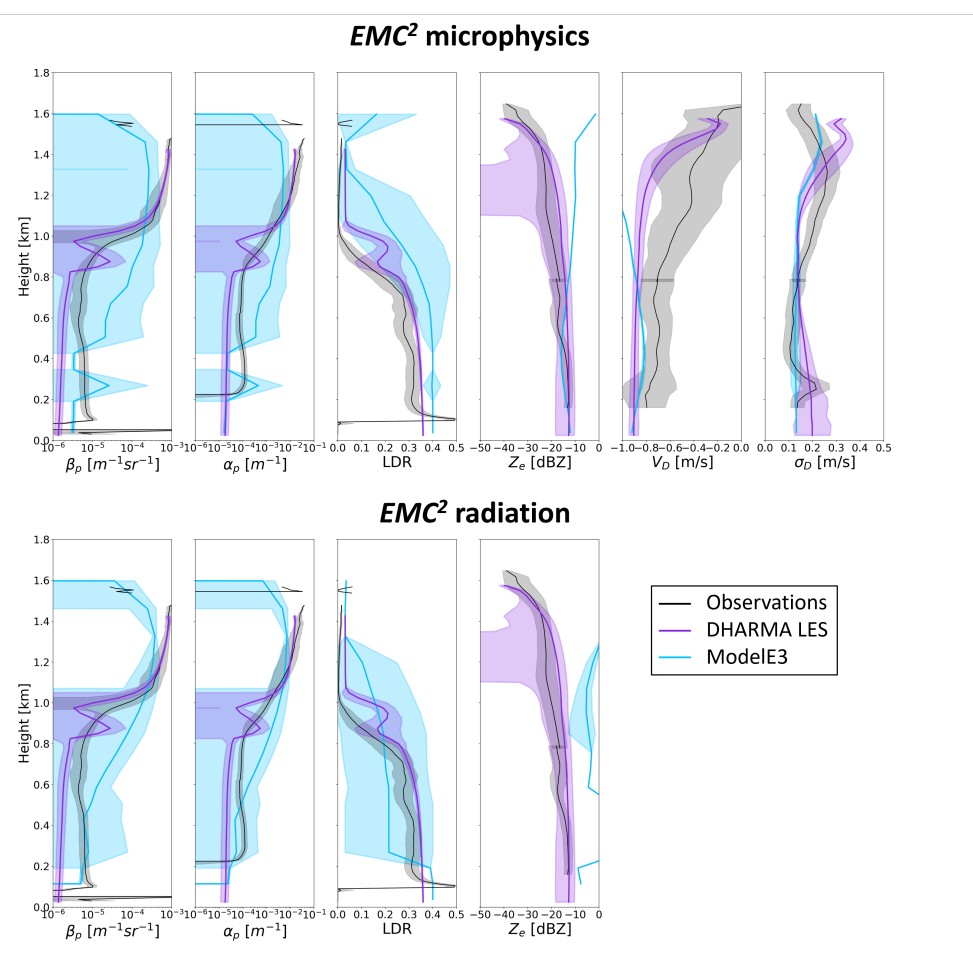

**Figure 6.** Mean profiles of observed (time-averaged) and simulated (subcolumn-averaged) HSRL and KAZR variables calculated using EMC²'s microphysics (top) and radiation (bottom) approaches as shown in figs. 4 and 5 (full lidar signal attenuation and radar SNR masks are applied prior to averaging). Shaded regions denote the mean $\pm 1$ standard deviation.

The EMC² lidar output also highlights a few model weaknesses. For example, using the microphysics approach, cloud base (at a height of ∼1.0 km) is highly variable and may extend nearly down to the surface, which is in contradiction to the observations, where cloud base variability is on the order of a few hundred meters throughout the depicted period (fig. 4). Using the radiation approach, on the other hand, breaks in hydrometeor cover are seen below the fully overcast layer, as a





result of the generalized hydrometeor fraction used in the radiative transfer calculations, which can be lower or higher than the associated hydrometeor class fraction (e.g., fig. 3, right). Lesser (greater) generalized hydrometeor fractions relative to the associated hydrometeor class fraction therefore implies greater (smaller) subcolumn bin mixing ratios (see eq. 1), and hence, enhanced (diminished) associated $\beta_{p_{hyd}}$ and $\alpha_{p_{hyd}}$ values. Thus, while the occurrence of low-level cloud water-bearing bins

produces full attenuation of the simulated lidar signal in some subcolumns using the microphysics approach (fig. 4), the smaller $\alpha_{p_{hyd}}$ ensuing from the larger low-level generalized cloud fraction relative to the cloud water fraction (see fig. 3, right) causes no low-level (or any level in this case) signal extinction when the radiation approach is used (see fig. 4). Note that in both the microphysics and radiation approaches the subcolumn representation of hydrometeor mass remains consistent with the model output variables, i.e., eq. 1 holds for each hydrometeor class.

While EMC[2]'s processing using the microphysics and radiation approaches is largely consistent with the observations, LES, and each other (figs. 4 and 6), the empirical approach gives less consistent results. The $\beta_{p_{tot}}$ and $\alpha_{p_{tot}}$ at heights where cloud water is the dominating hydrometeor are reasonably represented. However, the empirical $\beta_{p_{tot}}$ and $\alpha_{p_{tot}}$ in the lower-elevation ice-dominated regions are significantly smaller relative to the observations, whereas the resulting LDR values below the overcast cloud layer match the observations and LES slightly better. The $Z_{e_{tot,att}}$ calculated using the empirical approach

is underestimated at low levels (fig. 5). These deviations of the empirical approach relative to the microphysics and radiation approaches are strongly influenced by the deficient consideration of a given model physics, as well as the limited flexibility of the empirical parametrizations, which were specifically derived for certain geographical regions and/or conditions. Thus, while some assumptions made in the development of the empirical parameterizations may overlap with ModelE3 phyiscs in this particular case, others may not.

The microphysics and radiation approaches exhibit $Z_{e_{tot,att}}$ values that are too large, especially at higher levels (figs. 5 and 6). EMC[2]'s radar processing using the microphysics approach provides the $V_{D_{tot}}$ and $\sigma_{D_{tot}}$ variables in addition to the $Z_{e_{tot,att}}$ variable. As indicated from fig. 5, both of these calculated variables show grossly reasonable correspondence between the observations and the SCM, except at heights above ~0.8 km, where $V_{D_{tot}}$ values show large deviations (fig. 6). These deviations are mainly the result of relatively fast fall velocities (see Table 1) and the dominance of large snow hydrometeors

over $Z_{e_{tot,att}}$ at these levels (not shown).

    Fig. 7 delineates the three hydrometeor classification methods currently implemented in EMC[2] applied over the processed model output, with the COSP emulator "observing" the model domain from the top down, similar to the on-line simulator. Congruent with the description above of the calculated radar and lidar variables, the radar-sounding and modified fixed lidar threshold methods show the domination of liquid hydrometeor classes above ~1.0 km, the prevalence of precipitating hydrom-

eteors at lower levels (with lower occurrence using the radiation scheme approach), and the occasional full attenuation of the simulated lidar signal in the case of the lidar-based classification method. The COSP emulator detects a clear liquid-bearing subcolumn bin signal at cloud tops. However, because the cloud-top layer is highly reflective, generating large lidar scattering ratio values (the ratio of total to molecular attenuated backscatter), most of the underlying layers are either classified as "undefined" or generate signals too weak to be detected.





**Figure 7.** Hydrometeor classification of ModelE3 configuration Tun3 SCM output for 05:00 UTC processed using EMC$^2$'s microphysics (left) and radiation (middle) approaches while considering signal attenuation and detectability: (from top to bottom) radar-sounding method, modified fixed lidar variable threshold method, and the COSP lidar simulator emulator (top-down view). (Right) Brief summary of each classification method.

Using each of the three classification methods, phase ratio statistics can be generated with EMC$^2$ offering a method for analyzing the SCM simulation. Fig. 8 portrays the temporal evolution of the SCM simulation from the view of the simulated instruments and classification methods using the microphysics or radiation approaches. When radar-sounding and fixed lidar threshold methods are applied while using the microphysics approach, the evolution of the simulated cloud appears self-consistent between the two methods and generally follows the prototypical appearance of nearly continuously precipitating





liquid-bearing cloud layers with weakly varying cloud base height (e.g., de Boer et al., 2011; Fridlind and Ackerman, 2018; Silber et al., 2021a). Here, the radar-sounding skill is associated with the diminished cloud water fraction relative to the large fraction of the other hydrometeor classes (fig. 3, right) and the method's ability to correctly detect cloud water layers. The capabilities of this method should nonetheless be considered carefully when directly compared with observations because

realistic sounding profiles typically lack the fine temporal resolution emulated by EMC[2] here, and while in-situ observations can provide a robust characterization of liquid-bearing cloud layers, they can also produce sporadic false positive or negative cloud detections (e.g., Silber et al., 2020, fig S1; Vassel et al., 2019). In the case of the modified fixed lidar threshold method, the low-level phase ratio skill originates from the hydrometeor-bearing subcolumn bins being largely classified as "ice". This classification decision is the result of the low $\beta_{p_{tot}}$ and moderate-to-high LDR (e.g., fig. 6) produced by the prevalence of ice

hydrometeors relative to cloud water (fig. 3, right) and the greater mass (and likely volume due to the spherical representation) of these hydrometeors relative to rain (fig. 3, left).

The COSP emulator using the microphysics approach with a top-down view is consistent with the example in fig. 7, in which hydrometeor detection is limited by the optically-thick and highly reflective cloud-top region, resembling observational

retrievals (e.g., Cesana and Chepfer, 2013; Cesana et al., 2016).

Using the radiation approach, phase ratios more frequently show sharper transitions between the extreme values (all liquid or all non-liquid) stemming from the utilization of the generalized hydrometeor fraction. In the case of the radar-sounding method, for example, there is a distinct dominance of liquid-bearing bins, which only require any amount of cloud water to be classified as such. This dominance originates in the common occurrence of some cloud water mass mixing ratios in model grid

cells throughout the SCM simulation (fig. 3, left) combined with the implementation of the generalized hydrometeor fraction, which necessarily increases the overlap between cloud water and other hydrometeor classes. The limited number of model grid cells in which the subcolumn bins exhibit a more mixed behavior are the result of the randomized component of the subcolumn generator, which does not necessarily require overlap between cloud and precipitating hydrometeors (see sect. 2.1). Based on these classification results, we suggest that the radar-sounding method could lead to hydrometeor classification favoring

liquid-bearing classes when the radiation approach or similar model output with a generalized hydrometeor fraction is used.

The modified fixed lidar threshold method is relatively consistent with the microphysics approach (fig. 8), even though some times (mainly around 06:00 and 10:00 UTC) are characterized by greater (smaller) relative liquid occurrence at lower (higher) model levels (phase ratio values closer to 0.5). These phase ratio differences relative to the microphysics approach are the result of the convolution of the generalized hydrometeor fraction and its deviation relative to the cloud water fraction (fig. 3, right).

Using a top-down view, the COSP emulator agrees with both its application using the microphysics approach as well as the COSP output from the on-line simulator implemented in ModelE3, which utilizes the radiation approach (fig. 8. Unlike the on-line simulator, the emulator detects some "undefined" hydrometeors at low levels (down to the surface), which can be explained by the lack of model interpolation onto a uniform vertical grid and/or small differences in the compiled simulator code related to signal attenuation (e.g., the accumulation of optical thickness).

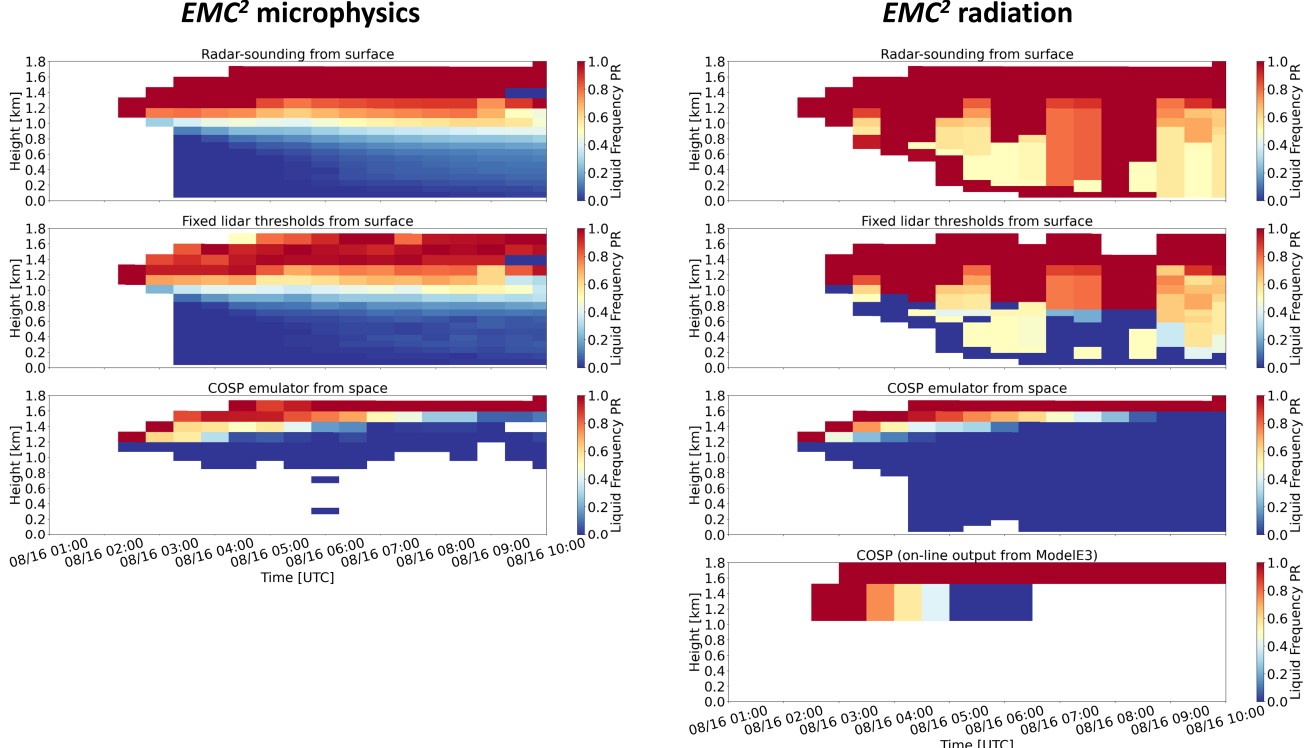

**Figure 8.** Time-height curtain plots showing the liquid-bearing frequency phase ratio (calculated over all hydrometeor-bearing subcolumn bins) of ModelE3 configuration Tun3 SCM simulation using the microphysics (left) and radiation (right) approaches. The classification methods used to calculate the phase ratios are (from top to bottom) the radar-sounding method, the modified fixed lidar variable threshold method, the COSP lidar simulator emulator (top-down view), and the on-line COSP lidar simulator implemented in ModelE3, which is only processed using ModelE3's radiation approach. In this figure, the "mixed" class of the radar-sounding method is counted as liquid, while the "undefined" classes in the other two methods are treated as "non-liquid" even though in some cases they are more likely to be liquid-bearing (e.g., the "undefined1" class in the modified fixed lidar variable threshold method).

Fig. 8 demonstrates the sensitivity of phase ratio statistics to the classification method, the viewing direction of the examined instrument, and the method by which "liquid" and "non-liquid" or "ice" classes are being counted. It shows that the use of forward simulators alone is not a guarantee for an "apples-to-apples" comparison, which requires matching processing steps to ensure its robustness.

## 5   Summary

EMC$^2$ provides an easy to use and flexible framework for the analysis of large-scale model output and its direct comparison with ground-based observations via the generation of subcolumns intended to explicitly represent a sub-grid scale variability,





and the simulation of ground-based (and air- or space-borne) radars and lidars. EMC$^2$'s framework is already tailored to the MG2 2-moment microphysics while using single-particle scattering LUTs and has the proper infrastructure required for it to be customized to other similar schemes, as well as high-resolution model output. EMC$^2$'s option for using radiation scheme logic in the subcolumn generator and simulator enables direct comparison with other on-line active instrument simulators (e.g.,

the COSP lidar simulator) with a bottom-up or top-down view option that can bridge between different methodologies by evaluating differences between the outputs resulting from their implementation.

Because it is generally designed to emulate ground-based systems, EMC$^2$ is suitable for the evaluation of column output extracted from global simulations against long-term ground-based datasets. The general adaptability of the software code to other climate models and instruments via the `model` and `instrument` Python classes renders EMC$^2$ as a flexible framework

to enable consistent and reproducible post-processing methods and evaluation across multiple models.

An AWARE case study was used to illustrate the application of EMC$^2$ to LES and SCM simulations of a highly super-cooled Antarctic cloud, including the utility of the program for hydrometeor classification using radar-sounding, lidar variable thresholds, and COSP emulator methods. The ModelE3 SCM using configuration Tun3 showed general agreement with the observations at the examined simulation time as well as with the baseline DHARMA LES used to develop this case study (see

Silber et al., 2019a). The LES output can be processed with EMC$^2$ after a few adaptations made only to an inherited `Model` class (see sect. 3). Thus, although it was developed for large-scale models, EMC$^2$ can also be used to compare cloud resolving or LES models with observations (as shown for DHARMA). EMC$^2$ also allows the implementation of advanced scattering model calculations in the forward calculations via customized LUTs that could be matched to some scattering assumptions made by models, for example, the implementation of the MODIS C6 calculations in both ModelE3's radiation scheme and

EMC$^2$.

The AWARE case study presented here is suitable for simulation by any global model in SCM mode (see input file repository specified under code and data availability). Case study observations, as well as ModelE3 SCM and DHARMA LES inputs and outputs from EMC$^2$ used to produce all examples above are also provided for step-by-step illustration (see code and data availability). We plan that additional case study examples will similarly be provided to illustrate results under differing cloud

regimes.

Planned future additions to EMC$^2$ include an extension to ground-based scanning radars, a Mie scattering calculator, spectral broadening estimates for the radar simulator, and a multiple-scattering model for the lidar simulator, all of which will be configured for consistency with model physics and output fields. We invite the community to take advantage of the framework provided by EMC$^2$ and to contribute to its further development and applications.

**Appendix A:  Lists of acronyms, abbreviations, and symbols**





**Table A1.** Acronyms and subscript abbreviations used in this manuscript.

| Name | Definition |
|---|---|
| ACT | Atmospheric Community Toolkit |
| ARM | Atmospheric Radiation Measurement user facility |
| ATB | attenuated total backscatter |
| AWARE | ARM West Antarctic Radiation Experiment |
| CALIOP | Cloud-Aerosol Lidar with Orthogonal Polarization |
| CERES | Cloud's and the Earth's Radiant Energy System |
| COSP | Cloud Feedback Model Intercomparison Project Observational Simulator Package |
| DHARMA | Distributed Hydrodynamic Aerosol and Radiative Modeling Application |
| EMC$^2$ | Earth Model Column Collaboratory |
| ESM | Earth System Model |
| GISS | Goddard Institute for Space Studies |
| HSRL | high spectral resolution lidar |
| IWP | ice water path (in $g/m^2$) |
| KAZR | Ka-band ARM zenith-pointing radar |
| LDR | linear depolarization ratio |
| LES | large eddy simulation |
| LUT | lookup table |
| LWC | liquid water content (in $g/m^3$) |
| LWP | liquid water path (in $g/m^2$) |
| MG2 | Gettelman and Morrison (2015) (2-moment microphysics scheme description) |
| MODIS | Moderate-Resolution Imaging Spectroradiometer |
| MPL | micropulse lidar |
| SCM | single column model |
| SNR | signal-to-noise ratio (in dB) |
| WACR | W-band ARM cloud radar |
| XSACR | X-band scanning ARM cloud radar |
| cl | cloud water (cloud liquid) |
| ci | cloud ice |
| pl | rain (precipitating liquid) |
| pi | snow (precipitating ice) |
| hyd | a hydrometeor class (cl, ci, pl, or pi) |
| tot | a total variable, incorporating multiple hydrometeor classes and/or cloud types (convective and/or stratiform) |
| att | attenuated (backscatter or radar reflectivity factor) |
| vol | volumetric (per unit volume) |
| gas | refers to the main atmospheric gases attenuating radar signals ($O_2$ and $H_2O$) |



**Table A2.** Symbols of variables and parameters used in this manuscript and their units (unless explicitly stated otherwise in the text).

| Symbol | Definition | Units |
|---|---|---|
| $z$ | height | meters |
| $q$ | mixing ratio | kg/kg |
| $w$ | verical velocity | m/s |
| $N$ | number concentration | $m^{-3}$ |
| $\rho_w$ | water density (1000 $kg/m^3$) | $kg/m^3$ |
| $\rho_b$ | bulk density of water or ice (1000 or 917 $kg/m^3$, respectively) | $kg/m^3$ |
| $\rho_a$ | air density | $kg/m^3$ |
| $\rho_{hyd}$ | density of a hydrometeor class | $kg/m^3$ |
| $\phi$ | particle size distribution | $m^{-4}$ |
| $r_e$ | effective radius | meters |
| $D$ | particle diameter | meters |
| $N_s$ | number of subcolumns | - |
| $f_{hyd}$ | fraction of a hydrometeor class | - |
| $f_{gen}$ | generalized hydrometeor fraction | - |
| $s, h, t$ | subcolumn (index), height (model level index), and time coordinates, respectively | - |
| $m_{hyd}$ | complex refractive index of an hydrometeor class | - |
| $K_m$ | $K_m = (m_{hyd}^2 - 1)/(m_{hyd}^2 + 2)$ | - |
| $\|K_w\|^2$ | dielectric factor for water | - |
| $\Phi_{hyd}$ | constant fluffiness factor of an ice hydrometeor class | - |
| $\lambda_l$ | lidar operating wavelength | meters |
| $A$ | geometric cross-section | $m^2$ |
| $Q_e$ | extinction efficiency | - |
| $Q_{bs}$ | backscattering efficiency | $sr^{-1}$ |
| $\beta_m$ | molecular backscatter cross-section | $m^{-1}sr^{-1}$ |
| $\beta_p$ | particulate backscatter cross-section | $m^{-1}sr^{-1}$ |
| $\alpha_p$ | particulate extinction cross-section | $m^{-1}$ |
| $T_m^2$ | two-way molecular transmittance | - |
| $\tau$ | accumulated optical thickness at a model level base (bottom-up view) or top (top-down view) | - |
| $\eta$ | multiple scattering coefficient | - |
| $\lambda_r$ | radar operating wavelength | meters |
| $Z_e$ | equivalent reflectivity factor | dBZ |
| $V_D$ | mean Doppler velocity | m/s |
| $\sigma_D$ | spectral width | m/s |
| $Z_{e_{min}}$ | minimum detectable $Z_e$ | dBZ |
| $Y$ | one-way integrated attenuation at a model level base (bottom-up view) or top (top-down view) | dB |



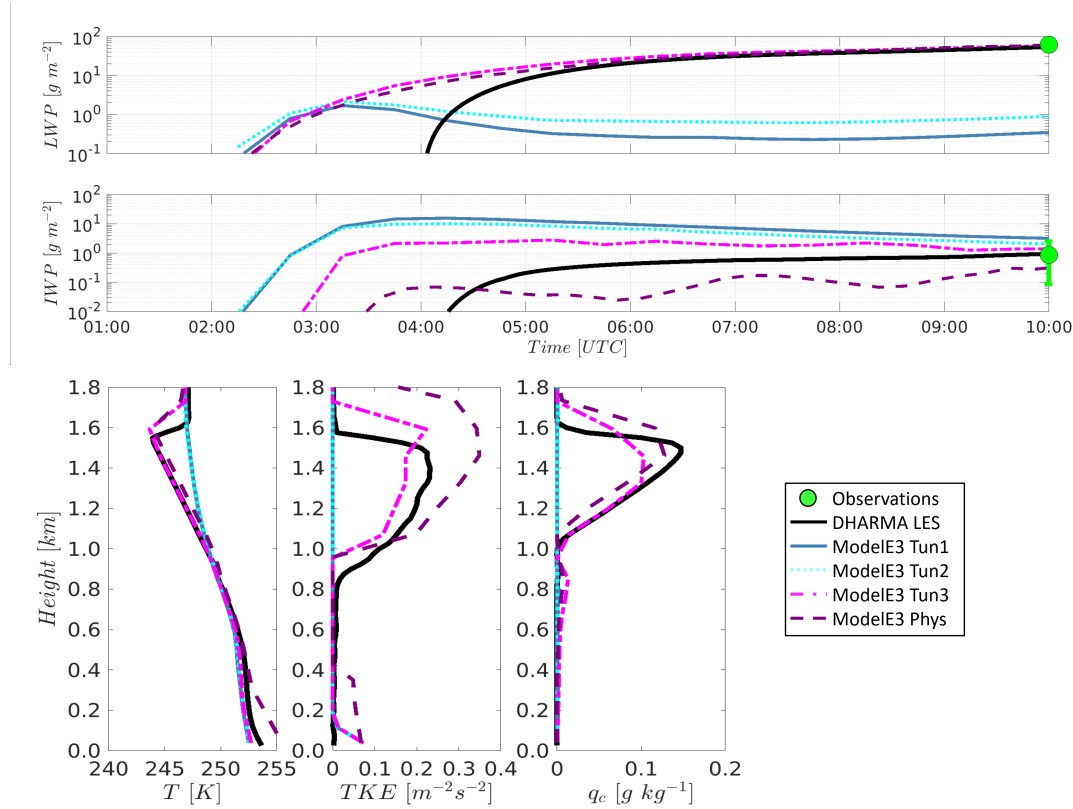

**Figure A1.** Time series showing (top) liquid water path (LWP) and (middle) ice water path (IWP) in each case study simulation using the DHARMA LES (baseline simulation in Silber et al., 2019a), and the four ModelE3 configurations (Tun1, Tun2, Tun3, and Phys). The Eulerian retrievals shown for 10:00 UTC (error bars denote uncertainty) correspond to the time at which the Lagrangian simulated domain approaches McMurdo Station (see sect. 4.2). (Bottom, from left to right) Temperature, turbulent kinetic energy, and cloud water mixing ratio profiles at the end of the SCM and DHARMA LES simulations.

## Appendix B: ModelE3 SCM Output Using the Four Different Model Configurations

Fig A1 illustrates time series of liquid water path (LWP) and ice water path (IWP) from the ModelE3 SCM case study output using configurations Tun1, Tun2, Tun3, and Phys (see sect. 4.3), as well as the temperature, turbulent kinetic energy (TKE) and cloud water mixing ratio profiles at the end of the simulation. Out of these four model configurations, only configurations Tun3 and Phys maintain substantial amounts of LWP in this highly supercooled cloud case study and agree with both the depicted DHARMA LES output and the observed LWP retrievals (see Silber et al., 2019a). Moreover, the SCM using each of these two configurations is able to develop a cloud-top inversion and TKE, both of which are driven by radiative cooling of cloud water, consistent with the DHARMA LES (see A1, bottom panels) and various polar cloud observations (e.g., Morrison et al., 2012; Silber et al., 2019a). All of ModelE3's configurations except for Tun1, which generates the largest amount of ice, are within



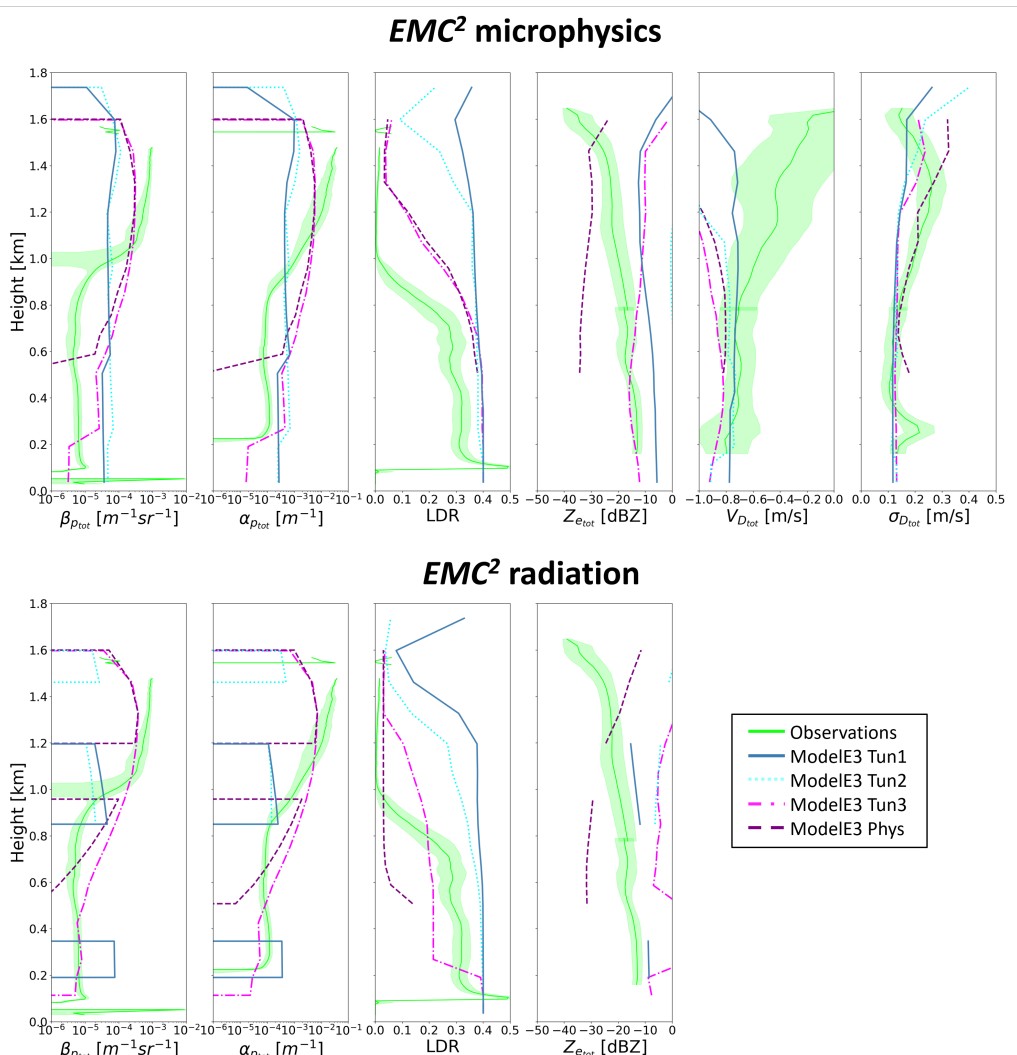

**Figure A2.** Mean profiles of observed (time-averaged) and simulated (subcolumn-averaged) HSRL and KAZR variables calculated using EMC$^2$ microphysics (top) and radiation (bottom) approaches using the four ModelE3 configurations (see legend), without applying a lidar and radar extinction or SNR masks, respectively.

the range of IWP estimated retrieval uncertainty. Out of those three model configurations, Tun3 is closest to the retrieved IWP and best matches the cloud formation and evolution in the LES.

Because the supercooled cloud is developed faster in the SCM than in the LES model (see A1, top and middle), we choose to demonstrate EMC$^2$ using the SCM output corresponding to 05:00 UTC. Fig A2 shows the mean lidar and radar variable profiles at that model time step from EMC$^2$ for each of the four model configurations together with the time-averaged observed profiles. Using either the microphysics or radiation approach, configurations Tun3 and Phys best match the observations. Most



of the radar and lidar variables calculated using these two configurations are largely consistent with each other, but overall, the Tun3 SCM output shows the best agreement with the mean observed profiles.

To summarize, two of ModelE3's final four configurations, that is, configurations Tun3 and Phys, show reasonable agreement with both the observed quantities as well as the LES output variables. Because configuration Tun3 performs slightly better, we

focus on this model configuration for detailed comparison with LES and observations.

*Code and data availability.* The most recent EMC$^2$ code is available on GitHub at https://github.com/columncolab/EMC2/. The EMC$^2$ Version 1.1 code described and used in this study together with KAZR and HSRL measurements, EMC$^2$-processed ModelE3 and DHARMA LES model output data files, and a Jupyter Notebook demonstrating the reproduction of the plots in this manuscript using EMC$^2$, are available at Zenodo (http://doi.org/10.5281/zenodo.5115252; Silber et al., 2021b). The SCM initialization files (sounding + forcing) required to run

the case study simulation are available at the Mendeley Data repository under http://dx.doi.org/10.17632/gz4gdn3jvz.1.

*Author contributions.* IS and AMF conceptualized the simulator and subcolumn generator with essential support from ASA. RCJ and SC conceptualized the open-source Python framework. IS and RCJ developed the EMC$^2$ code. ASA contributed to structuring general climate model logic and ModelE3's logic specifics. JV supported the consistency and conceptual implementation of the instrument logic. JD provided the Yang et al. (2013) and Ding et al. (2017) single-particle scattering LUTs and added essential information required to reproduce

and maintain the MODIS C6 consistency. IS prepared the manuscript with contributions from RCJ. All authors reviewed and edited the manuscript.

*Competing interests.* The authors declare that they have no conflicts of interest.

*Acknowledgements.* We thank Gregory Elsasser, Bastiaan van Diedenhoven, Maxwell Kelley, and Ed Eloranta for helpful discussions. IS is supported by DOE grants DE-SC0018046 and DE-SC0021004. SC and RJ were supported by the U.S. Department of Energy (DOE)

Atmospheric System Research (ASR), Office of Science, Office of Biological and Environmental Research (BER) program, under contract DE-AC02-06CH11357 awarded to Argonne National Laboratory. AMF and ASA were supported by the NASA Modeling Analysis and Prediction Program. Resources supporting this work were provided by the NASA Center for Climate Simulation (NCCS) at Goddard Space Flight Center.



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
