# Peer review of "The Earth Model Column Collaboratory (EMC2) v1.1: An Open-Source Ground-Based Lidar and Radar Instrument Simulator and Subcolumn Generator for Large-Scale Models"

_Geoscientific Model Development, 2021_

## Referee Comment (RC2)

[referee-annotated manuscript omitted]

---

## Author Comment (AC1)

**Author Responses**

We thank both reviewers for their valuable comments and helpful suggestions, which we think helped improve the manuscript. Our responses and revisions are enumerated below.

**Reviewer #1 (Comments to Author):**

This manuscript documents a new observation simulator for model applications, focused on ground based radar and lidar at individual locations. It documents the simulator methodology, and illustrates a case study application to mixed phase clouds over Antarctica. The paper is well written, and should be publishable with minor revisions. Specifically, there is quite a bit of qualitative analysis in figures 4-5 before getting quantitative in figure 6. There is also discussion of, but no demonstration of the phase ratio. In particular, estimating the phase ratio from lira and radar properties when the 'truth' for the model exists is a bit strange, and doesn't seem to be the point of a simulator. But at least illustrate the method, and how it might differ or reproduce model 'truth'. I also think the description could be tightened up in the introduction. But it is basically a good paper, suitable for GMD with these minor revisions, echoed in specific comments below.

We understand the reviewer's interest in the 'truth' for the model. The main problem with using the 'truth' (e.g., the distribution of condensate mass and related metrics) is that even if it is known for a given model output, it introduces an inverse problem (e.g., retrieving an atmospheric state variable from the observations), and hence, cannot be consistently evaluated using observations. Instrument simulators convert this model evaluation problem into a forward problem (generating observed variables using model output fields), which provides a consistent definition between different models and between models and observations, thereby enabling direct comparisons to be made (see also the discussion in Hillman et al., 2018).

We already emphasize the importance of making direct comparisons using similar methodologies and metrics at the end of sect. 3.3 (after discussing fig. 8 in detail):

"Fig. 8 demonstrates the sensitivity of phase ratio statistics to the classification method, the viewing direction of the examined instrument, and the method by which "liquid" and "non-liquid" or "ice" classes are being counted. It shows that the use of forward simulators alone is not a guarantee for an "apples-to-apples" comparison, which requires matching processing steps to ensure its robustness."

Figs. 4 and 5 are indeed qualitative, but we think that it is, nonetheless, important to show them as they enable the reader to perceive the ability of $EMC^2$ to produce forward-calculated variables comparable to observations. While it is also true that fig. 6 is more quantitative (no color scale), the general impression concerning a model's output behavior observed through the simulator transfer function is valuable, especially for potential users who wish to have the option of making such quick qualitative evaluations.

Our understanding is that in "no demonstration of the phase ratio", the reviewer refers to the mass phase ratio calculated using the raw model output. To address this comment, we added the mass phase ratio from the raw ModelE3 SCM output and wrote a paragraph discussing it per our arguments above:

"We note that the treatment of the COSP emulator's "undefined" subcolumn bins as "ice" to produce phase ratio statistics leaves the impression that only ice hydrometeors

exist below cloud top. However, a rather different impression of mostly liquid water dominance, though not as stark as in the radar-sounding method using the radiation approach, is perceived when the mass phase ratio calculated using the raw SCM output is examined (fig. 8, lower-left). Contrary to the COSP emulator, treating "undefined" bins as "ice" in the modified fixed lidar threshold method increases its apparent frequency phase ratio agreement with the mass phase ratio in multiple time-height bins. Phase classification depends on instrument measurement characteristics and limitations and hydrometeor properties such as their class, relative mixture with other hydrometeor classes, as well as their size distributions. Therefore, such an apparent agreement between different variables and phase occurrence metrics, as well as between the same variables and metrics based on different instruments and/or methodologies, should be taken with a grain of salt (cf. Cesana et al., 2021; see also Silber et al., 2021c)."

Specific Comments:

Page 1, L16: What is a cosp Lidar simulator emulator?

COSP is a package of multiple simulators, each imitating a different remote-sensing satellite instrument. One of these simulators is a lidar simulator imitating measurements made by the Cloud-Aerosol Lidar with Orthogonal Polarization (CALIOP) on-board the CALIPSO satellite. We didn't implement the actual COSP lidar simulator FORTRAN code, but instead added an emulator of that simulator; hence, the "COSP lidar simulator emulator".

Page 1, L17: So if Python it has to be run offline? Is it open source and publicly available? You never state that in the text. At least note it.

We note both explicitly and implicitly that EMC$^2$ is an open-source Python code in multiple locations in the text:

1. First, in the actual name of the package (Earth Model Column *Collaboratory*).
2. Abstract:
   "Here we present the Earth Model Column Collaboratory (EMC$^2$), *an open-source* ground-based lidar and radar instrument simulator and subcolumn generator…"
3. The final paragraph in the Introduction:
   "Here we present the Earth Model Column Collaboratory EMC$^2$, *an open-source* ground-based lidar and radar simulator and subcolumn generator."
4. Final sentence in the Summary section:
   "We invite the community to take advantage of the framework provided by EMC$^2$ and to contribute to its further development and applications."

We also mention that EMC$^2$ is currently an off-line simulator (sect. 2.4.2): "Noting that EMC$^2$ operates off-line, hydrometeor class fall velocities …"

Page 2, L19: Provide an example? Suzuki et al 2015 and/or Bodas-Salcedo et al 2013 maybe?

We added references for these two papers to the Introduction:

"Meaningful model evaluation benefits from a direct ("apples-to-apples") comparison with observations (e.g., Bodas-Salcedoet al., 2014; Suzuki et al., 2015)."

Page 2, L21: But retrieving microphysics is not the point. Microphysics in the model is used to simulate the observable (e.g. reflectivity pro backscatter or extinction).

We modified that sentence to make it clearer:

"However, model evaluation is challenging because of observational detectability constraints (e.g., signal extinction), and lack of retrievals or large uncertainties in some microphysical and atmospheric state quantities by these instruments, for example, hydrometeor number concentration or water content."

Page 4, L1: Note #1 and #2 for the rad and micro approaches.

We removed that sentence from the text as part of our response to the reviewer comment on Page 21, Figure 6, so this comment is no longer relevant.

Page 8, L6: Can you list all the tuning parameters?

Unfortunately, no, because this is not a ModelE3 descriptive paper or a user manual. Model tuning is discussed in the GISS-E2.1 article describing ModelE3's predecessor, but the tuning procedure and tuning parameters are quite different in the case of ModelE3. Several tens of ModelE3's tuning parameters will be provided to readers as part of an upcoming manuscript describing ModelE3's machine-learning tuning effort (Elsasser et al., in prep).

Page 11, L9: This is strange to simulate. The model knows explicitly what it's hydrometers are. Why try to approximate them from the radar/Lidar? Applying this to observations would be useful, but the point of a simulator is to cast the model in observation space…I assume you will compare outputs to model hydrometers l(truth) later?

This comment was largely addressed in our response to the general reviewer comment. We added text following the sentence noted by the reviewer to motivate using the classification masks in accordance with our response above:

"Once the total lidar and/or radar variables are calculated, EMC$^2$ can be used to classify the subcolumn simulator output. Classification masks can serve as tools for direct comparisons between the simulator output and observational data utilizing similar classification methodologies, some of which can be used to calculate water phase ratios…"

Page 12, L18: But what is the emulator? How is trained? Unclear what this COSP emulator is. Please add a few sentence description.

We added a few sentences describing the COSP emulator:

"The emulator of the COSP lidar simulator follows the same equations and logic of the on-line lidar simulator (Cesana and Chepfer, 2013) implemented in numerous climate models. In short, the attenuated total backscatter (ATB) calculated in the COSP emulator routine while assuming $\eta = 0.7$ is used to calculate the lidar scattering ratio (the ratio of total to molecular attenuated backscatter) for the detection of hydrometeors in subcolumns by selecting scattering ratio values larger than 5. Calculated cross-polar ATB as a function of the total ATB is then used to classify the detected hydrometeors into liquid or ice, based on an empirical phase discrimination line. As the last step of this classification method, hydrometeors below (top-down lidar view) or above (bottom-up lidar view) a subcolumn bin with scattering ratio larger than 30 are classified as "undefined"."

Note that an emulator, by its definition in the Oxford Advanced Learner's Dictionary, does not require training. We now clarify that at the first instance of the word emulator in the main text (sect. 2.5):

"… and the COSP lidar simulator emulator (henceforth referred to as the COSP emulator; 'emulator' is used here in its generic sense rather than a machine-learning context, and there is no training involved)."

Page 12, L30: Where is the model class with data in the figure?

We do not understand the question here. The model output data are loaded and stored in the Model class object as already shown in fig. 1's flowchart and noted in the text:

"Once loaded through the Model class internal methods, model output data are stored within the Model object …"

Page 14, L5: Would these model specific things be in the instrument class or model class? Seems like they should be in the model class? Please clarify.

That depends on whether the specifications are for instrument and measurement characteristics (Instrument class) or model output field names and cloud scheme logic (Model class). All of this information is already specified in the text above the sentence noted by the reviewer, for example:

"The Model class contains model output field namelists and default hydrometeor parameters (Table 1)."

"… the Instrument class contains relevant information about the instrument being simulated (some of which is listed in Table 2) as well as the single-particle and bulk scattering calculation LUTs …"

Page 17, L10: This is a good motivational sentence that I'm not sure was well reflected in the introduction, either to the whole paper or just section 4. Suggest stating this succinctly earlier.

That is an excellent suggestion. We revised the final paragraph of the Introduction:

"Here we present the Earth Model Column Collaboratory (EMC$^2$), an open-source ground-based lidar and radar simulator and subcolumn generator, which is designed to operate over large-scale model output while being faithful to the physics implemented in models' microphysics or radiation schemes but can also be applied to high-resolution model output. EMC$^2$ enables detailed evaluation of atmospheric thermodynamic profile and cloud properties extracted from local, regional, and global simulation outputs against long-term ground-based, air- or space-borne datasets."

Page 17, L22: Please state why you know it's the autoconversion parameterization at least.

We ran LES using the same model configuration except for the autoconversion parameterization, which was switched between Khairoutdinov and Kogan (2000) and Seifert and Beheng (2001). That is the reason we state "(not shown)" at the end of the previous sentence:

"This reduced amount of rainwater in the SCM simulation relative to rainwater dominance in the simulations of Silber et al. (2019a) is largely the result of the different autoconversion parameterization schemes implemented in ModelE3 (Seifert and Beheng, 2001) relative to that implemented by default in the DHARMA LES (Khairoutdinov and Kogan, 2000), which produces significantly smaller rain mass mixing ratios in this case (not shown), …"

Page 18, L3: Good agreement is not quantitative. See comment on figure 4: skip to figure 6 and use mean profiles to make the statement quantitative please.

We modified this sentence to tone it down a bit by using the adjective "apparent":

"The processed LES output exhibits good apparent agreement with the observations, …"

Page 19, Figure 4: what do the white lines represent in the EMC2output?

We suspect that the reviewer refers to the apparent white vertical line in each of the microphysics approach panels. Because we set the number of subcolumns to 100 and some of the hydrometeor class fractions are quite consistently closer to 0.99 (fig. 3), only 99 subcolumns are to be allocated with hydrometeors. Because $EMC^2$ uses the maximum-random overlap approach in allocating hydrometeors to subcolumn bins (i.e., hydrometeor clusters are extended vertically), the hydrometeor-free subcolumn in the resultant subcolumn snapshot shown in fig. 4 appears as if there is a white line in the plot.

Figure 6 is much better than Figure 4. Maybe showing one row to orient the reader. But then show Figure 6 and make discussion more quantitative.

We discussed the motivation to have both figs. 4 and 5 in addition to fig. 6 in our response to the general comment. We think that having both figs. 4 and 6 condensed into a single figure would make it much more difficult for readers to orient themselves in the condensed figure, and hence, we prefer keeping these two figures as is.

Page 19, L2: Again: reasonable is not quantitative.

We agree that the word reasonable is not quantitative. We reworded that sentence in a way similar to our response to the previous comment on Page 18, L3:

"Evaluation of heights with full cloud cover, indicated by the large $\beta_{p\_tot}$ and $\alpha_{p\_tot}$ values (see also fig. 3, right), suggests that the SCM has an apparent reasonable agreement with the observations there."

Page 20, Figure 5: Here having time mean profiles and temporal standard deviations would be more effective at discerning differences than the color scales you have chosen. Maybe just skip this figure and clean up figure 6?

We addressed this comment in our previous responses above.

Page 20, L7: appear to agree. Again, not quantitative.

That is correct, it is not quantitative but qualitative yet not definite, and therefore, we do not see a reason to further modify the text here.

Page 21, Figure 6 basically has all the information of Figure 4 and 5. Maybe show only one figure or one row of fugue 4 and discuss this figure instead. Add empirical approach? Would be nice to compare the model approaches on the same plot. Since only one line is different between top and bottom rows, I suggest you combine the plots to one for each variable, with all EMC2 methods on the same plot.

The empirical approach was originally implemented in $EMC^2$ to serve as a reference for evaluating the implementation of the radiation and microphysics approaches (whether the simulator output is "in the ballpark"; in other words, bug fixes). However, the empirical approach is deficient in its consideration of a given model physics as well as the limited flexibility of empirical parametrizations, which are generally derived for specific instrument operating wavelengths and certain geographical regions and/or conditions; hence, using this approach could impact comparisons as the simulator output is confounded by the utilized parameterizations. Therefore, after contemplating

the reviewer's comment and some of the comments received from the second reviewer, we eventually decided to omit the empirical approach from this article (that is, from the text and figures 4 and 5) and focus on the two main approaches faithful to model physics. We note that the empirical approach is turned off in $EMC^2$ by default, and hence, most likely won't be used by an end-user.

Page 22, L7: Which method and result is more correct relative to the observations?

We think that an answer to this question would be somewhat subjective, as it depends on which metric does one harness to evaluate the methods in use. Moreover, we need to be careful here because stating which method is "better" would be misleading, as the choice of which method to use should be based on research objectives and associated assumptions.

Page 22, L15: Why is empirical approach not in figure 6? Remind the reader if there is an already stated reason.

We addressed this comment as well in our response to the comment on Page 21, Figure 6.

Page 24, L12: What is the COSP emulator exactly? What is the method? It's not the same code as the on line COSP simulator, so how is COSP emulated? This is not explained.

We addressed this comment as well in our responses to the reviewer comments on Page 1, L16 and Page 12, L18.

Page 25, Figure 8: Why is the vertical resolution of the on-line COSP simulator different than the COSP emulator? Where do you get more vertical resolution?

The COSP simulator has a coarser vertical resolution than ModelE3's vertical spacing within the troposphere. We already note that in the text when describing the COSP simulator emulator in sect. 2.5:

"Note that, unlike the on-line COSP lidar simulator, this emulator operates using the model vertical levels and does not interpolate the model output onto an evenly-spaced vertical grid."

Page 25, L1: I'm not seeing phase ratio statistics in figure 8 as mentioned here. Please clarify what you mean. See general comment: showing the phase ratio using model truth and EMC2 processed data would be good.

We addressed this comment in our response to the general reviewer comment. We wish to emphasize that phase ratio is a rather general name for various metrics describing the occurrence of liquid or ice hydrometeors relative to one another (e.g., frequency phase ratio, mass phase ratio). The fact that different metrics are occasionally treated as comparable suggests that cross-evaluation of different phase ratio results from the literature should be made with caution (see also Cesana et al., 2021, and the discussion in Silber et al., 2021).

Appendix B is never really referred to in the text (looking at different model configurations) and should be moved to the main text with the figure (should be B1, not A2) added to the main text (maybe remove figure 4 & 5).

We thank the reviewer for noticing that the figures for Appendix B were tagged as if they belonged to Appendix A. This error is corrected now. To keep the case study

example concise, we prefer to keep the discussion on the ModelE3 configurations in Appendix B.

We note that Appendix B is already referred to in the second sentence of sect. 3.3:

"The SCM using this configuration … produced the best agreement with the observations and the LES relative to the other three configurations (see Appendix B)."

---

## Author Comment (AC2)

**Author Responses**

We thank both reviewers for their valuable comments and helpful suggestions, which we think helped improve the manuscript. Our responses and revisions are enumerated below.

**Reviewer #2 (Comments to Author):**

The authors have done a nice job in creating a radar-lidar simulator, and I applaud them for making this code freely available and taking the time to document such here. I have no major concerns about the material per se, but there are some aspects of the description that I do not understand very well.

We thank Roj Marchand for the feedback and for thoroughly reading the manuscript. Indeed, a leading objective of this project is to create a freely available simulator with a relatively accessible code, which would enable consistent model evaluations and comparisons. We have made additional clarifications in the text per our responses below.

Comments:

I found the description of the overlap scheme difficult to follow and did not come away with a clear understanding of how cloud and precipitation are overlapped. This is important. I hope you will excuse if I point you some work that I and Ben Hillman have done on this topic (https://doi.org/10.1029/2017JD027680). In particular, I don't understand how f_gen is created or how this resulted in the structure shown in column 4 of Figure 4. Perhaps it would help if you included some examples in the overlap description highlighting key aspects of this approach.

We understand the confusion in the text. $EMC^2$ does not calculate $f_{gen}$ but uses it if it is used in a model's radiation scheme and is a model output field. In the case of ModelE3, for example, $f_{gen}$ corresponds to cloud fraction where the layer-average cloud opacity exceeds that of precipitation, or vice versa. The $f_{gen}$ field depicted in Fig. 3 of the manuscript (lower-right panel) appears as a combination of the cloud and precipitation fraction fields at different times and heights (see upper-right panels in Fig. 3). Because our use of the "the overlap rule" between water phases was confusing, we omitted that term from the text and reworded the relevant sentences. The errors associated with the maximum-random overlap found by Hillman et al. (2018) are interesting, and we now refer to them in the text.

We revised the text in sect. 2.1 to address this comment:

[revised manuscript text omitted]

p. 2 l. 8 – "The representation of cloud processes in models is continuously advancing" - Perhaps specific the type of model. Certainly the references do not include all types of atmospheric models?

Added "large-scale" to be consistent with the references.

p. 2 l. 12 – "This variability results from, among other sources, model weaknesses concerning atmospheric processes such as cloud geometrical and optical thicknesses…" - "procecess" such as geometrical and optical thickness. These two things are not "processes" and I am not sure if any of the other things listed after this are truely processes, either. Perhaps change to "... concerning their ability to predict ... "

We adopted the reviewer's recommendation:

"This variability results from, among other sources, model weaknesses concerning their ability to predict atmospheric state variables and processes such as …".

p. 2 l. 19 – remove "in these profiling cases, performing"

Done.

p. 2 l. 20 – "… model evaluation is challenging because of observational detectability constraints (e.g., signal extinction), and lack of retrievals of some microphysical …" - perhaps add ".. or large uncertainties in some ... ".

Done.

p. 2 l. 30 – "Because of the demanding computation associated with the emulation of satellite measurements, COSP is typically implemented on-line into models' code to facilitate output." - I am not sure I think this is the dominant reasons such is done. Rather I suspect that having to save/output detailed information that are inputs to the simulators in major factor. Perhaps rephrase to note there are a variey of reasons and perhaps include the above reason, as well?

We agree. The sentence was modified:

"Among a variety of reasons such as the demanding computation associated with the emulation of satellite measurements and the ability to use and output detailed data to and from simulators, COSP is typically implemented on-line into models' code to facilitate output."

p. 3 l. 2 – remove "largely"

Done.

p. 3 l. 4 – "Statistics calculated using multiple generated subcolumns (faithful to the processed model's physics) can be directly compared with the associated observations…" - perhaps "(which are nominally faithful to the model's physics)". As I am sure you appreaciate this is not easily done, and I fear that subcolumns are typically not entirely consistent. For example, subcolumns would nominally represent the overlap of cloud and precipitation droplets and represent spatial variability in (the existance / amont of) precipitaiton below cloud base. But evaporation schemes in coarse models often do not consider this variability in a way that is consistent with what is used in the radar and lidar simulator.

We agree with adding "nominally". Sentence reworded:

"Statistics calculated using multiple generated subcolumns, which are nominally faithful to the processed model's physics, can be directly compared with the associated observations, thereby mitigating spatial resolution biases and errors."

p. 4 l. 1 – "The bespoke radiation and microphysics approaches …" - On first reading, I found it confusing to have the word "bespoke" here and think it would be simpler to just use "radiation and microphysics approaches".

After reading through some of the first reviewer's comments and contemplating some of the comments from this reviewer, we decided to omit the empirical approach from this manuscript (that is, from the text and figures 4 and 5) and focus on the two main approaches faithful to model physics. The empirical approach was originally implemented in EMC$^2$ to serve as a reference for evaluating the implementation of the radiation and microphysics approaches (whether the simulator output is "in the

ballpark"; in other words, bug fixes). However, the empirical approach is deficient in its consideration of a given model physics as well as the limited flexibility of empirical parametrizations, which are generally derived for specific instrument operating wavelengths and certain geographical regions and/or conditions; hence, using this approach could impact comparisons as the simulator output is confounded by the utilized parameterizations. The sentence mentioned by the reviewer was removed from the text as part of these revisions.

p. 4 l. 20 – "or a generalized hydrometeor fraction (fgen) used in the model radiation scheme …" - From the later figures, I see this is important, and I think you need to explain how this is calculated.

This comment was addressed in our response to the general reviewer comment.

p. 4 l. 28 – "Stratiform cloud hydrometeors are allocated to subcolumn bins unoccupied by convective cloud hydrometeors while noting that cl and ci are allocated simultaneously to consistently follow the overlap rule." - I am not sure I undestand what this means.   Forgetting overlap between vertical layers for the moment, if a given layer has 30% stratoform cl and 10% stratoform cl 10%, does column with ci also have cl (i.e. there is 10% mixed phase and 20% liquid only and 0% ice only) ?

That is correct. Our revisions to the text in response to the general reviewer comment address this comment as well.

p. 5 l. 3 – "are allocated to subcolumns without convective-stratiform no-overlap restrictions, …" -Does "without ... no overlap restrictions" mean?   Perhaps rephrase to make meaning clearer, and try to remove double negative.

That is a good idea. The text now reads:

"Convective and stratiform precipitating hydrometeors (pl and pi) are allocated to subcolumns without convective-stratiform restrictions, such that convective and stratiform precipitation may co-exist in a single subcolumn bin."

p. 5 l. 4 – "…  i.e., convective and stratiform precipitation may co-exist in a single subcolumn bin while complying with the overlap rule." - I am not sure I understand this.

I think this means that if stratiform pl > stratiform cl, the "additional" pl (pl-cl) are placed only in clear columns that are below an overlying column with startiform cl or ci (randomly) or if stratiform cl+ci in the overlying later is too smal then below an overlying clear column (precipitation somehow associated with no cloud) but never below or in a column with convective cl or ci ??

Does the same hold true for convective pl > convective cl?

How are pl and pi located with respect to each other?

It might be helpful to have some example cases here so readers can visualize how these rule interact.

We think that our revisions to the text in response to the general reviewer comment, in which we added some general examples and removed the "overlap rule" term address this comment.

p. 5 l. 26 – "… EMC2 utilizes qhyd and Nhyd to calculate the hydrometeor size distribution, φhyd(D, s, h, t), fully consistent with the MG2 scheme" - So where does

the D dependence come from ?     I presume the point is the the MG2 scheme has a distribution defined by ONLY two parameters.   You should explain this more clearly.

Revised the text to:

"In the microphysics approach, applicable only to stratiform hydrometeors, per hydrometeor diameter D, EMC$^2$ calculates the hydrometeor size distribution, $\varphi_{hyd}$(D,s,h,t), defined by $q_{hyd}$ and $N_{hyd}$, fully consistent with the MG2 scheme (see Morrison et al., 2009, eq. 1-3)."

p. 6 l. 13 – "The total αp and βp (αptot (s, h, t) and βptot (s, h, t), respectively) are calculated as the sum of each of these variables for cl, ci, pl, and pi." - It is confusing to write on line 25, at the top of this subsection, that this only applies to cl and then later it appears that you are applying this approach to all hydrometeor types.   I know where this is going, but I think most readers are going to be confused.

We think that the confusion originated from the use of "stratiform clouds" instead of "stratiform hydrometeors" in the first sentence of this subsection (see the quote in our response to the previous comment).

p. 7 l. 5 – "EMC2 assumes by default that the lidar signal is extinct at a level where τtot = 4." - I am not sure this is a good idea.   The signal will get vanishly small on it own, so why artifically truncate it ?

First, we note that EMC$^2$ does not artificially truncate the simulated lidar signal, but simply generates an extinction mask (as shown in Figure 4) that can be used by an end-user. We now clarify that in the text:

"Lidar signal extinction at visible wavelengths typically occurs at an optical thickness of 3-5 (e.g., Sokolowsky et al., 2020, fig. 4), and hence, EMC$^2$ assumes by default that the lidar signal is extinct at a level where $\tau_{tot} = 4$ to produce a lidar signal extinction mask."

Concerning the actual extinction level, we agree that the attenuated backscatter signal would become very small on its own because of the dependence on the cumulative extinction along the lidar signal path (eq. 4 in the text). Therefore, if one wishes to compare simulator output to elastic lidar measurements (such as in the case of an MPL, ceilometer, or CALIOP), then comparing the attenuated backscatter signal might suffice. However, in the case of HSRLs and Raman lidars enabling the direct retrieval of both particulate extinction and backscatter cross-section (all of which are particulate properties that are independent of the cumulative extinction along the lidar path – see eq. 2 in the text), the extinction level in the simulator needs to be determined to allow a direct ("apples-to-apples") comparison (as shown in Fig. 4).

Now, the question is at which level does the signal become fully extinct? The answer in the case of real observations is subjective and depends on the instrument in use. When using elastic lidar measurements, the lidar scattering ratio (total attenuated to molecular backscatter ratio) can be used to determine full extinction by setting a fixed threshold. In the case of HSRLs and Raman lidars, particulate optical thickness and/or molecular backscatter SNR can be used to determine full extinction, again, using a fixed threshold (note that when the signal is vanishingly small, the particulate extinction and backscatter cross-section retrievals would often not converge anyway).

We decided to implement the optical thickness threshold to produce the EMC$^2$ extinction mask because it is independent of multiple scattering effects, unlike the scattering ratio. Inclusion of multiple scattering coefficient (η) values smaller than 1

would significantly impact the calculated attenuated backscatter ($\beta_{att}$), and hence, also the scattering ratio, in a non-linear fashion, depending on the cumulative optical thickness along the lidar signal path ($\tau$). In other words, $\eta$ introduces a strong dependence on the hydrometeor scenes generated by the model. The plot below shows how $\beta_{att}$ is enhanced when considering $\eta$ in its calculation as a function of $\tau$ for different $\eta$ values. In the extreme case of $\eta = 0.7$, a value that is often used in the processing of CALIPSO satellite data (e.g., Winker, 2003), tenuous clouds with a total optical thickness of 1 could enhance $\beta_{att}$ by ~80% but the impact of $\eta$ could increase by an order of magnitude even below an optical thickness of 4. In the case of ground-based lidars, the field-of-view is typically much smaller compared to the CALIOP spaceborne lidar, and therefore, the value of $\eta$ is likely much bigger and closer to 1, making the actual impact of $\eta$ much smaller relative to the case of CALIOP. Moreover, $\eta$ effectively serves as a simplification of the rather complex $n^{th}$ order scattering of photons by hydrometeors (e.g., Eloranta, 1998), which is heavily dependent on hydrometeor PSDs. Therefore, implementing a default fixed $\eta$ value other than 1 would introduce a confounding factor that is not faithful to model output. For this reason, we plan to add a multiple scattering model that will use the PSDs resolved by the model as input. That said, one should keep in mind that some processed lidar datastreams already include multiple scattering treatments such that comparisons between such observations and simulator output that includes multiple scattering considerations might introduce over-correction biases.

[Figure]

To summarize, given the small FOV of ground-based (and airborne) lidars, which would be most relevant for EMC$^2$, an $\eta$ value of 1 prevents introducing additional confounding factors, maintains faithfulness to model physics (until a consistent multiple scattering model would be added to the simulator), and therefore, enables the most consistent comparison with observations possible. (Note that as in the case of other variables and parameters mentioned in the manuscript, $\eta$ is an optional input parameter to EMC$^2$, so different values can be specified for processing).

We added some text (final paragraph of sect. 2.3.1) discussing the value of $\eta$:

"where … $\eta$ is the multiple scattering coefficient. The value of $\eta$ is set to 1 by default, effectively implying no multiple scattering by hydrometeors or perfect multiple scattering treatment. Such an assumption is not realistic in the vast majority of cases even though effective $\eta$ values closer to 1 are more likely given the common narrow field-of-view of ground-based (and airborne) lidars. While a value of 1 is most likely unrealistic, it precludes the introduction of additional confounding factors stemming

from the dependence of multiple scattering effects on the hydrometeor particle size distributions (e.g., Eloranta, 1998), which could significantly impact model-observation comparisons of $\beta_{att,tot}$ (not shown). Moreover, this dependence of multiple scattering on particle size distributions suggests that a fixed $\eta$ value smaller than 1 would impact the faithfulness of the lidar simulator to the model physics. That said, we note that $\eta$ can be manually set to other fixed values based on the physical assumptions made or certain empirical results (e.g., Winker, 2003), and that the determination of the extinction level based on $\tau_{tot}$ is independent of $\eta$."

p. 7 l. 14 – "and $\eta$ is the multiple scattering coefficient, the value of which is assumed to be equal to 1 by default" - This is a signficant limitation. Perhaps add some dicussion to how this effects the utility of the simulated fields!

Our response above addresses this comment.

p. 7 l. 23 – "rehyd is the effective radius of a hydrometeor class in the model grid cell" - I don't follow. How was this specified?

$r_{ehyd}$ is a model output field. Now we mention that in the text:

"and $r_{ehyd}$ is the effective radius of a hydrometeor class in the model grid cell, provided as a model output field."

p. 8 l. 5 – "In the case of ModelE3, for example, $\Phi_{hyd}$ is set by default to an intermediate value of 0.5, and generally serves as one of many tuning parameters" - So if I understand the density of cloud-ice and cloud-snow is set to be constant in this ModelE3, and doesn't depend on particle size?

I am not sure I think this is a good idea and is certainly not very general.

That is correct, and to our knowledge, all current-generation climate models (ModelE3, CESM2, E3SM, etc.) use that assumption. Note that the bulk LUTs used by the radiation schemes of these models often implicitly include varying size-dependent density assumptions (e.g., following Platnick et al., 2016 for ModelE3; Neale et al., 2012, ch. 4.9.4 for CESM2 and E3SM), but these assumptions are not embedded in the microphysics (in fact, that was one the incentives to add both radiation and microphysics approaches to EMC$^2$).

p. 8 l. 13 – "In order to calculate the Qe,volhyd and Qbs,volhyd LUTs, we assumed the same gamma distribution parameters as those implemented in the C6 dataset (see Hansen, 1971, eq. 1), consistent with the bulk LUTs utilized by ModelE3's radiation scheme. Following the calculations of _phyd and _phyd , the total variables αptot and βptot as well as τhyd, _tot, and βptot,att are calculated as in the microphysics approach (sect. 2.2.1)." - Similar to an comment earlier, there is still a dependence on the shape of the distribution burried in the LUTs, which if I understand is set in the radiation code??

(A) I think you should explain more clearly what the assumptions are.

As mentioned in our response to the previous comment, the PSD and shape assumptions are only manifested in the LUTs, and we already mention the shape assumptions used by the C6 collection in the same paragraph highlighted by the reviewer:

"The default $Q_{e,vol\_hyd}$ and $Q_{bs,vol\_hyd}$ … implemented in EMC$^2$ were calculated using … single-particle scattering LUTs for a severely roughened 8-column ice aggregate (Yang et al., 2013) in the case of solid hydrometeors. These ice aggregate scattering

calculations have been shown by Holz et al. (2016) to enable a closure between infrared Moderate-Resolution Imaging Spectroradiometer (MODIS; Platnick et al., 2003) and visible Cloud-Aerosol Lidar with Orthogonal Polarization (CALIOP; Winker et al., 2009) satellite ice optical thickness retrievals, and were included in the MODIS collection 6 (C6) cloud product (Platnick et al., 2016)."

(B) Also, I am concerned that your simulator is not very general in this regard. If my model makes a different set of assumptions regarding the shape of the size distribution, what am I supposed to do? Does the EMC code allow me to generate a new set of look up tables ?

Indeed, the source and assumptions embedded in bulk LUTs used by various models need to be scrutinized to produce bulk LUTs for instrument operating wavelengths that can be used with EMC$^2$, and thereby maintain simulator consistency. We see no shortcuts here, but this downside highlights one of the strengths of an open-source collaboratory, and that is the option for users/developers of certain models to add relevant information that could be easily implemented in EMC$^2$ and used by other community members. Some open-source Python packages allow generating LUTs for certain ice particle shape assumptions (e.g., SnowScatt; see Ori et al., 2021) but the implementation of these assumptions in the scattering calculation would not necessarily conform with the implementation of the shape assumptions in the target models. That said, coupling such packages to EMC$^2$ might be a valuable addition in the future.

p. 8 l. 30 – "Using the resultant Zehyd , VDhyd is then calculated by implementing the hydrometeor class terminal velocities parametrization used in the MG2 scheme (cf. Morrison and Gettelman, 2008, Table 2)." - I presume one can edit the values used in table? Is anything other than a power-law allowed ?

One can indeed edit the table values. As we now note in the software description subsection:

"Thus, the various parameters mentioned in the next subsections (e.g., all parameters shown in Tables 1 and 2) as well as the LUTs, can be easily specified and set to match configurations and assumptions implemented in different large-scale models …"

Only power-law parameterizations are currently implemented in the simulator, but since the power-laws are manifested in a single line of code, options for different parameterization types can be easily plugged into the simulator code if necessary.

p. 8 l. 31 – "In the calculation of VDhyd we neglect the model grid cell vertical wind, …" - at coarse resolution this might be OK but not at scales typical of radar measurements that the subcolumns nominally represent! This seems like a major limitation that deserves some discussion. What should a user do or NOT do with the simulated velocities ?

While it is true that subcolumns nominally represent smaller scales, they should be examined statistically (e.g., as shown in fig. 6). We already note that in the introduction:

"Statistics calculated using multiple generated subcolumns, which are nominally faithful to the processed model's physics, can be directly compared with the associated observations, thereby mitigating spatial resolution biases and errors",

as well as in the subcolumn generator subsection:

"Prior to the radar and lidar forward calculations, EMC$^2$ generates subcolumns for each model output column. These subcolumns emulate a higher model spatial resolution, which partially reconciles the locality of ground-based measurements and allows a more robust *statistical* model evaluation",

and the case study demonstration:

"When evaluating the processed model output against the observations, we essentially exchange temporal resolution with spatial resolution (three-dimensional model domain in the case of the LES) or an emulated spatial resolution (in the case of the SCM)."

Given that current large-scale models do not provide/implement a parameterization or a range of possible sub-grid air motion but only grid-cell averaged values, it would be wrong to add the grid-cell averaged vertical air motion and claim that $V_D$ in specific subcolumns (representing smaller scales) can be compared with observations, especially we when also consider the typically larger time steps in large-scale models.

We added some text to the sentence noted by the reviewer:

"In the calculation of $V_{D\_hyd}$ we neglect the model grid cell vertical wind, *w*, which predominantly has little impact on the $V_{D\_hyd}$ value, especially at coarser spatial and temporal resolutions typical to large-scale models."

Personally, I would not recomment using the velocities on small scales (i.e. you need to look at domain averages or something such that the air motions are arguably small).

We agree. As we quoted above from the case study example subsection, we should effectively exchange temporal resolution (time-averaged observations) with spatial resolution (subcolumn-averaged or domain-averaged model output).

p. 10 l. 7 – "Noting that EMC2 operates off-line, hydrometeor class fall velocities are typically reported in model outputs as weighted means." - I don't understand what you mean here.

In some cloud schemes such as ModelE3's convective scheme (see Elsasser et al., 2017), the mass-weighted fall velocities reported in model outputs cannot be used to derive the hydrometeor fall velocities as a function of diameter required to robustly calculate higher radar moments offline. We added some text to this bullet:

"Noting that EMC$^2$ operates off-line, hydrometeor class fall velocities are typically reported in model outputs as weighted means. Because not all cloud schemes enable back-tracing of hydrometeor class fall velocities as a function of particle diameter using analytical expressions and weighted output fields (e.g., the convective cloud scheme in ModelE3; see Elsasser et al., 2017), hydrometeor class fall velocities per subcolumn bin cannot be straightforwardly reproduced."

p. 11 l. 9 – "Once the total lidar and/or radar variables are calculated, EMC2 can be used to classify the subcolumn simulator output. " - Yes, but perhaps you could explain why you are doing this. What is the real-apple that is going to be compared to this simiulated-apple ? The word "classification" has appeared in this document only one-time prior to this point and not in a way that explains the purpose of this material.

Good point. We added text after this first subsection sentence:

"Once the total lidar and/or radar variables are calculated, EMC$^2$ can be used to classify the subcolumn simulator output. Classification masks can serve as tools for direct comparisons between the simulator output and observational data utilizing similar classification methodologies, some of which can be used to calculate water phase ratios."

We also added the word "direct" to the following sentence when discussing the modified fixed lidar variable threshold method:

"… the modified fixed threshold routine, which largely agrees with existing measurements yet acknowledges both model and observational uncertainties may allow better direct comparisons to be made."

p. 11 l. 26 – "By default, however, EMC2 includes two additional "undefined" classes that cover intermediate regions in the LDR-βptot, …" - Perhaps include a figure that illustrates the domain? ... in retrospect I see you did this in the figure in section 4. perhaps point readers to this inset here.

We already refer at the end of this sentence to the appropriate subsection containing the relevant illustration:

"By default, however, EMC$^2$ includes two additional "undefined" classes … (see sect. 3.3 for discussion and illustration of the default thresholds)."

p. 12 l. 14 – "The emulator of the COSP lidar simulator follows the same equations and logic of the on-line lidar simulator (Cesana and Chepfer, 2013) implemented in numerous climate models." - Summarize briefly ?

We added a summary to the text:

"The emulator of the COSP lidar simulator follows the same equations and logic of the on-line lidar simulator (Cesana and Chepfer, 2013) implemented in numerous climate models. In short, the attenuated total backscatter (ATB) calculated in the COSP emulator routine while assuming η = 0.7 is used to calculate the lidar scattering ratio (the ratio of total to molecular attenuated backscatter) for the detection of hydrometeors in subcolumns by selecting scattering ratio values larger than 5. Calculated cross-polar ATB as a function of the total ATB is then used to classify the detected hydrometeors into liquid or ice, based on an empirical phase discrimination line. As the last step of this classification method, hydrometeors below (top-down lidar view) or above (bottom-up lidar view) a subcolumn bin with scattering ratio larger than 30 are classified as "undefined"."

p. 12 l. 21 – "EMC2 depicts a workflow …" - You might consider reorganizing the material to give this description of the overall flow of the code at the start of section 2. It was obvious to me how all of this was going to fit together, but I can easily imagine many readers would benefit from seeing the big picture first.

That is an excellent suggestion, which we adopted (software description, now subsection 2.1, comes before the detailed description of the simulator and subcolumn generator in subsections 2.2 onward).

p. 13 l. 2 – "Currently, zenith-pointing instrument properties and scattering calculation LUTs are available for various lidars and radars …" - Does the code provide any facilitate user to generate LUT for other instruments ?

In v. 1.1 described in the text, the answer is no (refer to our response above to the comment on p. 8 l. 13).

p. 14 l. 2-5 – "Thus, the various variables … as well as the LUTs, can be easily specified and set to match configurations and assumptions implemented in different large-scale models, as well as complex scattering models more commonly implemented in cloud-resolving and LES models." - Is there a way to easily bypass the subcolumn generator?   If I have LES scale output or my own subcolumn generator can I use it with the rest of EMC2?     If yes, perhaps put this into the flow chart.

Setting the number of subcolumns to 1 practically skips the subcolumn generator processing. We updated the flowchart in fig. 1 and modified the text related to the flowchart:

"The Model object is then input to the subcolumn generator (sect. 2.2). Note that subcolumn generator processing can be practically skipped by setting the number of subcolumns ($N_s$) to 1."

p. 14 l. 17 – "EMC2 incorporates a suite of unit tests for each function … If the unit test passes with the developer's changes to the code, then the changes are approved to be a part of EMC2." – Very nice!

Thank you!

p. 18 Fig. 3 caption – "The generalized hydrometeor fraction …" - I do not understand what this means (or how it is calculated).   This is not really explained in section 2.   I presume the four panels above are giving f_hyd for each of the hydrometeor classes.   I understand how f_hyd is used in the simulator but not how the "generalized" fraction is used in the simulator (or for that matter in the radiation code).

Our response to the general comment addresses this comment as well.

p. 18 l. 5 – "A multi-layer cloud structure developed by the LES is suggested by the intermittent breaks in the large βptot and αptot values." - I not sure I follow, it seems rather the cloud ice field (Figure 3, third panel on the left) shows two layers what might be two layers but it seem to be embedded in a liquid layer ?

Indeed, the cloud ice embedded in cloud water shows a multi-layer structure based on mixing ratio (left panels in fig. 3). The cloud water mixing ratio also shows such a structure, though it is more difficult to discern in the logarithmic color scale depicted in fig. 3. Examination of $\beta_{p\_tot}$ and $\alpha_{p\_tot}$ calculated from the LES output per hydrometeor class (left and right panels below, respectively; pink – cl, green – pl, tan – ci, gold – pi, purple – total after applying a lidar extinction mask) indicates that the apparent multi-layer structure originates in cloud water rather than ice, mainly due to the much greater backscatter cross-sections. We modified the text accordingly:

"A multi-layer cloud water structure developed by the LES is suggested by the intermittent breaks in the large $\beta_{p\_tot}$ and $\alpha_{p\_tot}$ values, supported by a separate analysis of $\beta_{p\_hyd}$ and $\alpha_{p\_hyd}$ (not shown)."

[Figure]

p. 18 l. 8 – "Using the microphysics approach, the SCM sub-grid variability is more pronounced relative to the radiation approach…" - Please rephrase more precisely. I am not 100% sure what variability you mean here. I presume the variation in the yellow and orange colors in the top two rows is due to the variability in cloud water which is not considered in the radiation approach.

We rephrased this sentence:

"Using the microphysics approach, the SCM sub-grid variability is more pronounced relative to the radiation approach owing to the implementation of cloud water sub-grid variability (as defined in the MG2 microphysics scheme), which is not considered in the radiation approach."

p. 19 Fig. 4 caption – "(without the subcolumn generator)…" - What are the domian columns here? Are all the LES columns sequenced in time or stacked "x by y" in some way? Is this why there are periodic looking structure in the DHARMA results ?

That is correct. This is not an LES column sequence in time (as we already note in the caption: "The DHARMA LES output corresponds to 10:00 UTC …"), but the x by y dimensions being stacked onto the "domain column" dimension. We changed that part of the caption to:

"… DHARMA LES three-dimensional output (horizontal dimensions stacked onto the domain column dimension) processed using EMC$^2$ microphysics with the subcolumn generator turned off ($N_s = 1$), …"

p. 19 Fig. 4 caption – "EMC2 output using the radiation approach, …" - I do not nderstand how this structure emerged. Somehow the fractional converaged at 0.2 km appears to be 100 % but near 0% for all classes at 0.4 km ??

Recall that the radiation approach utilizes $f_{gen}$, the profile of which at 05:00 UTC (lower-right panel in fig. 3) is consistent with the radiation approach output depicted in fig. 4 (note that the logarithmic color scale in fig. 3 can be somewhat confusing in this case).

p. 19 Fig. 4 caption – "A full lidar signal attenuation mask of $\tau tot > 4$ ($\tau tot$ is the total accumulated optical thickness) is appled …" - I mis-understood what you mean my "appling a mask". To me "masking" means removing values wher t_tot < 4. Here you are identifying where such is the case. I suggest you rephrase a bit, and say "a mask in generated which identifies regions where ... "

We rephrased this figure caption sentence:

"A mask denoting full lidar signal attenuation generated using a total accumulated optical thickness ($\tau_{tot}$) condition of $\tau_{tot} > 4$ is plotted over the simulated data (hatched areas)."

p. 21 Fig 6 - Why not show the emprical profiles? You do discuss such.

As we explained in our response to the reviewer comment on p. 4 l. 1, we removed the empirical approach from the manuscript so this comment is no longer relevant.

p. 22 l. 7 – "Note that in both the microphysics and radiation approaches the subcolumn representation of hydrometeor mass remains consistent with the model output variables, i.e., eq. 1 holds for each hydrometeor class." - f_gen doesn't appear in equation 1 ... how does this work ?

As we already note in sect. 2.2, "… $f_{hyd}$ is the volume fraction of a hydrometeor class (e.g., $f_{cl}$ or $f_{pi}$) or a generalized hydrometeor fraction ($f_{gen}$) used in the model radiation scheme, at the same model level and time step."

Thus, when the radiation code is used, $f_{gen}$ is simply used together with $q_{hyd}$ to conserve mass following eq. 1.

p. 22 l. 20 – "The microphysics and radiation approaches exhibit Zetot,att values that are too large, especially at higher levels (figs. 5 and 6)." - So what does this mean as regards the model microphysics? (I think your objective here should be not simply to describe the difference, but provide the users some guidance on how to interpret differences).

We already describe the reason for these deviations at the end of that paragraph:

"The microphysics and radiation approaches exhibit $Z_{e\_tot,att}$ values that are too large … these deviations are … mainly the result of relatively fast fall velocities … and the dominance of large snow hydrometeors over $Z_{e\_tot,att}$ at these levels…"

p. 22 l. 22 – "As indicated from fig. 5, both of these calculated variables show grossly reasonable correspondence between the observations and the SCM…" - I am not sure I agree with this. If I understand, the observed Doppler velocity shown here include air motions not accounted for in the simulation and this needs to be discussed. Is the offset between the mean of the obs and simulation due to air motions or indicative of particle sizes that are too large (fall velocities that are too large)?

We agree with the reviewer that vertical motion could be an important factor impacting the comparison, especially if it was made using observations merely corresponding to a much shorter period of up to several minutes or so. However, over the course of an hour, a time-averaged profile serves as a filter leaving mainly longperiod waves (or apparent long-period waves, the result of a superposition of polychromatic oscillations) as the source of un-filtered vertical air motion. During the depicted case study (10:00-11:00 UTC), long-period waves could have been responsible for a vertical air motion of up to several cm/s (see fig. 2 in Silber et al., 2019), much smaller than the SCM deviations from the observations. Thus, air motion representation is not the leading source of the SCM deviations in this case, though it can be the leading source in other cases.

We modified the final sentence of this paragraph to reflect this discussion:

"A separate analysis (not shown) suggests that, in this case, these deviations are mainly the result of relatively fast fall velocities (see Table 1) and the dominance of large snow hydrometeors over $Z_{e\_tot,att}$ at these levels and not the product of vertical air motion being convolved into the radar moments only in the observations."

Likewise, the observed spectra width include turbulent broadening, which if I understand has been entirely neglected in the simulator. I think this needs to be made clear. How should I interpret the comparison. What does it mean if the simulated mean value is larger or smaller than the observed value?   What does it mean if the sigma widith is much larger than observed ?

That is a good point. The observed spectral width includes not only turbulent broadening but also shear broadening as well as spectral broadening due to the finite radar beam width. Without the implementation of all spectral broadening terms, the calculated $\sigma_D$ serves as a lower bound when compared with the observed values. We now clarify that in the text:

"We note that because spectral broadening terms other than the microphysical broadening are currently not considered in EMC[2], the calculated $\sigma_{D\_tot}$ generally serves as a lower bound for comparison purposes, i.e., the simulated $\sigma_{D\_tot}$ values need to be smaller to some extent than the observed values. As indicated from fig. 5, both $\sigma_{D\_tot}$ and $\sigma_{D\_tot}$ show…"

p. 23 Fig. 7 caption – "modified fixed lidar variable threshold method, …" - So there is no assement or adjustment of thresholds for multiple scattering ?   I think this would cause serious problems if you were looking down.

The typically enhanced multiple scattering in the top-down view is indeed influenced by the larger droplets near cloud top, but it is mainly driven by the FOV of the lidar instrument in use (i.e., CALIOP in most cases), so an airborne HSRL, for example, still suffers much less from multiple due to its significantly smaller FOV. We addressed this comment as well in our response to the reviewer comment in p. 7 l. 5, but we wish to emphasize again (as noted in the text) that the multiple scattering coefficient can be specified by the user, and that the COSP emulator does use a default η value of 0.7 as commonly assumed in the CALIPSO satellite community.

p. 25 fig. 8 caption – "In this figure, the "mixed" class of the radar-sounding method is counted as liquid, while the "undefined" classes in the other two methods are treated as "non-liquid" …" - I think this is rather misleading as it makes it look like the COSP emulator knows where ice is located, but it is really almost all undefined below the liquid top.

I would urge you to adopt a new scale with red colors being liquid fraction and blue colors being ice fraction ... or something that gives a more complete picture of the

situation. I think the reality is that phase much below cloud top is not known from the lidar signal for these liquid topped clouds.

This comment touches the main purpose (as we see it) of fig. 8 as already stated in the final paragraph of this subsection:

"Fig. 8 demonstrates the sensitivity of phase ratio statistics to the classification method, the viewing direction of the examined instrument, and the method by which "liquid" and "non-liquid" or "ice" classes are being counted. It shows that the use of forward simulators alone is not a guarantee for an "apples-to-apples" comparison, which requires matching processing steps to ensure its robustness."

We added to this figure a panel showing the mass phase ratio from the raw model output and added a paragraph further discussing this issue (before the final subsection paragraph quoted above):

"We note that the treatment of the COSP emulator's "undefined" subcolumn bins as "ice" to produce phase ratio statistics leaves the impression that only ice hydrometeors exist below cloud top. However, a rather different impression of mostly liquid water dominance, though not as stark as in the radar-sounding method using the radiation approach, is perceived when the mass phase ratio calculated using the raw SCM output is examined (fig. 8, lower-left). Contrary to the COSP emulator, treating "undefined" bins as "ice" in the modified fixed lidar threshold method increases its apparent frequency phase ratio agreement with the mass phase ratio in multiple time-height bins. Phase classification depends on instrument measurement characteristics and limitations and hydrometeor properties such as their class, relative mixture with other hydrometeor classes, as well as their size distributions. Therefore, such an apparent agreement between different variables and phase occurrence metrics, as well as between the same variables and metrics based on different instruments and/or methodologies, should be taken with a grain of salt (cf. Cesana et al., 2021; see also Silber et al., 2021c)."

p. 26 l. 26 – "… a Mie scattering calculator …" - I presume this would allow construction of LUT for spherical particles?   Any thoughts about non-spherical particles ?

Concerning spherical particles, that is correct. With regards to non-spherical (ice) particles, an optional avenue would be to plug a different Python package such as snowScatt to EMC$^2$. However, as noted above, we see no shortcuts with regards to ice shape assumptions because the SSRGA approach, for example, might not necessarily satisfy ice shape assumptions made in some models, which would require a more careful analysis and/or processing to produce appropriate LUT that would enable direct comparisons to be made.

p. 26 l. 27 – "… and a multiple-scattering model for the lidar simulator…" - This seem like a major limitation of the current design.   In particular, this seem very important to the entire classification activity.

We addressed this comment as well in our response to the reviewer comment in p. 7 l. 5.